# Revision of the Late Jurassic crocodyliform *Alligatorellus*, and evidence for allopatric speciation driving high diversity in western European atoposaurids

Jonathan P. Tennant and Philip D. Mannion

Department of Earth Science and Engineering, Imperial College London, London, UK

Corresponding author
Jonathan P. Tennant,
jonathan.tennant10@imperial.ac.uk

## ABSTRACT

Atoposaurid crocodyliforms represent an important faunal component of Late Jurassic to Early Cretaceous Laurasian semi-aquatic to terrestrial ecosystems, with numerous spatiotemporally contemporaneous atoposaurids known from western Europe. In particular, the Late Jurassic of France and Germany records evidence for high diversity and possible sympatric atoposaurid species belonging to *Alligatorellus*, *Alligatorium* and *Atoposaurus*. However, atoposaurid taxonomy has received little attention, and many species are in need of revision. As such, this potentially high European diversity within a narrow spatiotemporal range might be a taxonomic artefact. Here we provide a taxonomic and anatomical revision of the Late Jurassic atoposaurid *Alligatorellus*. Initially described as *A. beaumonti* from the Kimmeridgian of Cerin, eastern France, additional material from the Tithonian of Solnhofen, south-eastern Germany, was subsequently referred to this species, with the two occurrences differentiated as *A. beaumonti beaumonti* and *A. beaumonti bavaricus*, respectively. We provide a revised diagnosis for the genus *Alligatorellus*, and note a number of anatomical differences between the French and German specimens, including osteoderm morphology and the configuration and pattern of sculpting of cranial elements. Consequently, we restrict the name *Alligatorellus beaumonti* to include only the French remains, and raise the rank of the German material to a distinct species: *Alligatorellus bavaricus*. A new diagnosis is provided for both species, and we suggest that a recently referred specimen from a coeval German locality cannot be conclusively referred to *Alligatorellus*. Although it has previously been suggested that *Alligatorellus*, *Alligatorium* and *Atoposaurus* might represent a single growth series of one species, we find no conclusive evidence to support this proposal, and provide a number of morphological differences to distinguish these three taxa that appear to be independent of ontogeny. Consequently, we interpret high atoposaurid diversity in the Late Jurassic island archipelago of western Europe as a genuine biological signal, with closely related species of *Alligatorellus*, *Alligatorium* and *Atoposaurus* in both French and German basins providing evidence for allopatric speciation, potentially driven by fluctuating highstand sea levels.

## INTRODUCTION

Atoposaurids comprise a clade of small-bodied terrestrial and semi-aquatic crocodyliforms (*Owen, 1879*; *Joffe, 1967*; *Buscalioni & Sanz, 1990a*; *Thies, Windolf & Mudroch, 1997*; *Lauprasert et al., 2011*). Historically, they were considered to be the sister group to Eusuchia (*Joffe, 1967*; *Buffetaut, 1982*), but are now recovered in all phylogenetic analyses as the basal-most members of Neosuchia, which includes crown group crocodylians (*Benton & Clark, 1988*; *Buscalioni & Sanz, 1990b*; *Salisbury et al., 2006*; *Brochu et al., 2009*; *Pol & Gasparini, 2009*; *Adams, 2013*; *Sertich & O'Connor, 2014*). Atoposaurids were an important component of a range of Late Jurassic to Early Cretaceous western European ecosystems (Fig. 1), with less common occurrences extending their known stratigraphic range from the Middle Jurassic to the end-Cretaceous (168.3–66 million years ago [Ma]; Fig. 2) (*Owen, 1879*; *Buscalioni & Sanz, 1984*; *Buscalioni & Sanz, 1987a*; *Salisbury, 2002*; *Martin, Rabi & Csiki, 2010*; *Salisbury & Naish, 2011*). There is tentative evidence to suggest that atoposaurids might have persisted beyond the Cretaceous/Paleogene boundary, based on fragmentary material from the Middle Eocene of the Republic of Yemen (*Stevens et al., 2013*). The earliest known atoposaurid specimens are *Theriosuchus*-like teeth from the early (*Kriwet, Rauhut & Gloy, 1997*) and middle (*Knoll et al., 2013*) Bathonian (late Middle Jurassic) of southern France and the Bathonian of the UK (*Evans & Milner, 1994*), with *Theriosuchus sympiestodon* from the Maastrichtian of Romania the last known occurrence (*Martin, Rabi & Csiki, 2010*; *Martin et al., 2014*). Other putative and fragmentary occurrences potentially extend the distribution of Atoposauridae into the Late Jurassic–Early Cretaceous of Asia (*Young, 1961*; *Efimov, 1976*; *Wu, Brinkman & Lu, 1994*; *Wu, Sues & Brinkman, 1996*; *Wu, Brinkman & Lu, 1994*; *Storrs & Efimov, 2000*; *Cuny et al., 2010*; *Wings et al., 2010*) and North America (*Gilmore, 1926*; *Cifelli et al., 1999*; *Eaton et al., 1999*; *Fiorillo, 1999*; *Rogers, 2003*), and a late Early Cretaceous occurrence, *Brillanceausuchus babouriensis*, from Cameroon, might represent evidence for the presence of the clade in Gondwana (*Michard et al., 1990*).

Despite this research history and range of recent discoveries, there is currently little species-level taxonomic clarity or consensus on atoposaurid inter-relationships (*Owen, 1878*; *Owen, 1879*; *Wellnhofer, 1971*; *Buffetaut, 1982*; *Benton & Clark, 1988*; *Buscalioni & Sanz, 1988*; *Brinkmann, 1989*; *Brinkmann, 1992*; *Wu, Sues & Brinkman, 1996*; *Schwarz & Salisbury, 2005*). *Steel (1973)* considered Atoposauridae to comprise *Alligatorellus*, *Alligatorium*, *Atoposaurus*, *Hoplosuchus*, *Shantungosuchus*, and *Theriosuchus*. More recently, *Lauprasert et al. (2011)* recognised only four valid genera, *Alligatorellus*, *Alligatorium*, *Montsecosuchus*, and *Theriosuchus*, with the latter genus comprising four species: *T. grandinaris*, *T. guimarotae*, *T. ibericus* and *T. pusillus*. *Martin, Rabi & Csiki (2010)* augmented this species list with their description of *T. sympiestodon* from the Maastrichtian of Romania. *Schwarz-Wings et al. (2011)* followed this taxonomic scheme,

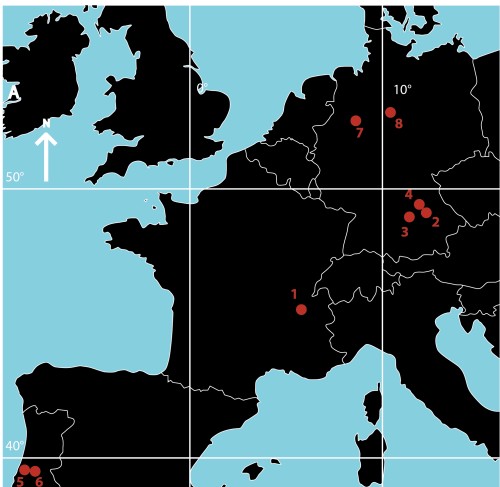
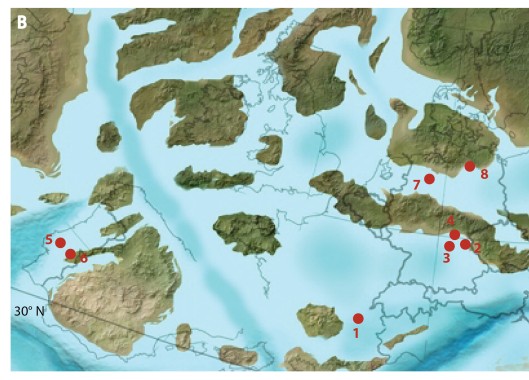

**Figure 1** (A) Geographic distribution of Late Jurassic atoposaurid specimen localities. 1, Cerin; 2, Kelheim; 3, Painten; 4, Solnhofen; 5, Guimarota; 6, Andrès; 7, Langenberg; 8, Uppen. Note that those localities not mentioned in the text all include occurrences of indeterminate remains of *Theriosuchus*; (B) Approximate palaeogeographic distribution of Late Jurassic atoposaurids. Map reconstruction from Ron Blakey, Colorado Plateau Geosystems, Arizona, USA (http://cpgeosystems.com/paleomaps.html).

but also regarded *Atoposaurus* as a valid genus, comprising the two species *A. jourdani* and *A. oberndorferi*. There are three currently recognised species of *Alligatorium*: *A. meyeri* from France (*Vidal, 1915*) and *A. franconicum* (*Ammon, 1906*) and *A. paintenense* (*Kuhn, 1961*; originally described by *Broili (1931)* as a possible occurrence of *A. franconicum*) from Germany. However, specimens of the latter two were lost or destroyed during World War II. An Early Cretaceous Spanish species originally placed in *Alligatorium* has since been assigned to a new genus, *Montsecosuchus* (*Vidal, 1915*; *Peybernes & Oertli, 1972*; *Buscalioni & Sanz, 1988*; *Buscalioni & Sanz, 1990a*).

*Gervais (1871)* originally erected the species name *Alligatorellus beaumonti* for two specimens from the Late Jurassic of Cerin, in eastern France. *Wellnhofer (1971)* later assigned these specimens to the subspecies *A. beaumonti beaumonti*, in recognition of differences between coeval specimens from Eichstätt, southeast Germany, for which he erected the subspecies *A. beaumonti bavaricus*. Both the French and German specimens have been regarded as *Alligatorellus beaumonti* by subsequent workers (e.g., *Buscalioni & Sanz, 1988*; *Schwarz-Wings et al., 2011*). As a result of these factors, the species-level composition and relationships of *Alligatorellus*, *Alligatorium* and *Atoposaurus*, as well as the Spanish *Montsecosuchus depereti*, remains poorly understood. This in part reflects a paucity of specimens, but also the flattened mode of preservation of the specimens concerned, which often obscures much of their morphology (*Meyer, 1850*; *Meyer, 1851*; *Gervais, 1871*; *Wellnhofer, 1971*; *Buscalioni & Sanz, 1990a*). This taphonomic signature results from their exclusive occurrence in lithographic limestones. Furthermore, *Theriosuchus* appears to have become a 'waste-basket taxon' for recently discovered small, basal neosuchian specimens from Asia and Europe. Unlike *Alligatorellus*, *Alligatorium*, *Atoposaurus* and *Montsecosuchus* which occur in lagoonal settings, *Theriosuchus* occurs in

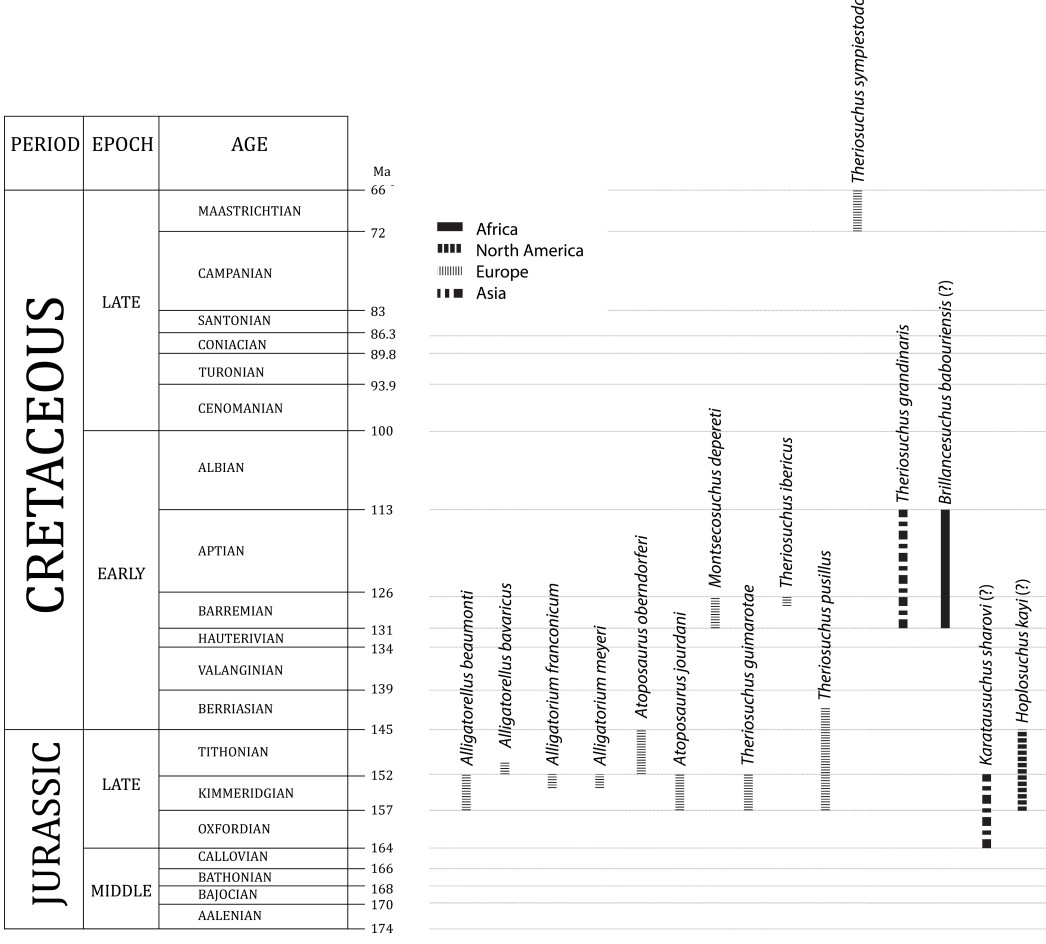

**Figure 2** Stratigraphic (including uncertainty) and geographic ranges of known and putative (denoted with a "?") atoposaurid species.

a range of transitional brackish onshore or near-shore environments (*Schwarz & Salisbury, 2005*; *Lauprasert et al., 2011*).

Given such potentially high European atoposaurid diversity within a narrow geographic and temporal range, and a lack of taxonomic consensus, a full revision of atoposaurid systematics is overdue. Presented here is a re-assessment of specimens of *Alligatorellus* from the Late Jurassic of France and Germany in the first of a series of papers in which we will revise the taxonomy, systematics and phylogenetic relationships of Atoposauridae. We refer the German occurrence to a new species of *Alligatorellus*, providing a comprehensive re-description, and make detailed comparisons with the French type species. We also consider the taxonomic affinities of an additional German specimen described as *Alligatorellus* sp. (*Schwarz-Wings et al., 2011*), and examine the osteoderm morphology of *Alligatorellus*, investigating its utility in atoposaurid systematics. Finally, we examine the taxonomy and validity of the contemporaneous, multispecific taxa *Alligatorium* and *Atoposaurus*, and discuss the diverse atoposaurid faunal composition of the Late Jurassic of western Europe.

## SYSTEMATIC PALAEONTOLOGY

Crocodylomorpha *Walker, 1970*
Crocodyliformes *Hay, 1930*
Mesoeucrocodylia *Whetstone & Whybrow, 1983*
Neosuchia *Benton & Clark, 1988*
Atoposauridae *Gervais, 1871*
*Alligatorellus* *Gervais, 1871*

*Note on taxonomy*: *Gervais (1871)* did not designate a holotype specimen in his original description of *Alligatorellus beaumonti*. *Wellnhofer (1971)* elected MNHN 15639 as the holotype of *A. beaumonti beaumonti*. As this is one of the two specimens described by *Gervais (1871)*, we follow *Wellnhofer (1971)* in considering MNHN 15639 to be the holotype for the genus and type species of *Alligatorellus beaumonti*.

*Wellnhofer (1971, p. 144)* provided the following diagnosis of *Alligatorellus* (translation adapted from *Schwarz-Wings et al., 2011*): (1) a large-sized atoposaurid (420–550 mm) with an acute-triangular skull and large orbits; (2) the supratemporal fossae are not internally fenestrated, and are connected to the orbit by a superficial furrow; (3) the nasal aperture is divided; (4) the tail is longer than half of the precaudal body length; (5) presence of a biserial osteoderm shield from the nuchal to the caudal region; (6) single osteoderms are sculpted; (7) presence of a lateral keel on the nuchal and dorsal osteoderms, whereas the caudal osteoderms bear a more medial keel; (8) ventral armour possesses two rows of scutes in the tail region; (9) the ventral scutes are oval and medially keeled.

*Comments:* In light of more recent atoposaurid discoveries and an improved understanding of their anatomy, much of *Wellnhofer*'s (*1971*) diagnosis requires revision. The first putative defining characteristic (1) is a feature that also describes the sizes of *Alligatorium meyeri*, *A. franconicum*, *Montsecosuchus depereti*, and *Theriosuchus pusillus*, and may in fact be an over-estimation of their size. The lack of internal fenestration (2) of the supratemporal fenestra is not seen in other atoposaurids, including *Alligatorium*, *Montsecosuchus*, and *Theriosuchus*, and is thus retained as a locally diagnostic feature. The division of the nasal aperture (3) is not visible in LMU 1937 I 26 as a result of crushing of the anterior-most portion of the snout, but is present in MNHN 15639. Regardless, this appears to be a feature shared by other atoposaurids including *Theriosuchus pusillus* (NHMUK PV OR48330) and *Theriosuchus grandinaris* (*Lauprasert et al., 2011*). The relative length of the tail (4) is a feature seen in other atoposaurids including *Atoposaurus* and *Theriosuchus pusillus* and appears to be widespread among Atoposauridae, as are characters (5) and (6). Indeed, osteoderm sculpting and a biserial osteodermal shield are present in *Alligatorium*, *Montsecosuchus*, and *Theriosuchus*. The presence, prominence, and position of a dorsal keel on the biserial osteoderms might be diagnostic at the generic level (7), although there are differences between the German and French specimens, as discussed below. The presence of a dual row of ventral osteoderms in the caudal region is also questionable (8), especially with respect to their morphology (9)—they are rarely and poorly preserved in the ventral region in both French and German specimens. It is probable that

post-mortem flattening has re-arranged the paravertebral dorsal osteoderms, which, when viewed laterally, might easily be misinterpreted as belonging to a ventral series. Finally, it should be noted that in the referred specimen of *A. beaumonti* (MNHN 15638), the osteoderms are much less visible, with just a single noticeable row overlying the anterior caudal vertebrae, and possibly a single row concealed underneath the dorsal vertebrae.

*Revised diagnosis:* Among currently recognised atoposaurids, *Alligatorellus* can be diagnosed based on the following unique combination of features and autapomorphies (highlighted with an asterisk): (1) rostrum unsculpted or substantially less so than cranial table; (2) cranial sculpting comprised of homogeneous shallow pitting; (3) absence of hypertrophied maxillary tooth 5, with homodont pseudocaniniform dentition; (4) frontal width between the orbits narrower than maximal width of nasals; (5*) broad frontal anterior process, not constricted; (6) absence of raised orbital or supratemporal rims; (7) unperforated supratemporal fenestra; (8*) anterior process of squamosal extends to the orbital margin; (9*) posterodorsal margin of parietals and squamosals completely covers dorsal occipital region; (10) smooth mandibular outer surface; (11) proportionally short first metatarsal; (12) dorsal surface of dorsal osteoderms completely sculpted, with parallel and straight anterior and posterior margins; (13*) dorsal osteoderms with longitudinal ridge along entire lateral margin; (14) caudal osteoderms with smooth, non-serrated edges.

A. beaumonti *Gervais, 1871*
A. beaumonti beaumonti *Wellnhofer, 1971*

*Holotype:* MNHN 15639, part and counterpart slabs preserving a near-complete, articulated skull and skeleton, missing the distal forelimb elements and part of the left hindlimb (Fig. 3).

*Referred specimen*: MNHN 15638, part slab comprising a near-complete articulated skeleton, missing the distal-most caudal vertebrae and part of the left forelimb (Fig. 4).

*Locality and stratigraphic age:* Cerin, Ain, eastern France; Kimmeridgian (Late Jurassic) (*Wellnhofer, 1971*).

*Preservation of holotype:* The specimen is dorsolaterally flattened and, on the part, the dorsal surface of the skull is embedded into matrix comprising grey lithographic limestone. This obscures both the lateral and ventral surfaces, and much of the mandible. Thirteen maxillary teeth are preserved. The complete, articulated axial skeleton is preserved, with the exception of the three posterior-most caudal vertebrae, and is overlain by a continuous sheath of parasagittal biserial osteoderms. At least eleven ribs are preserved *in situ* on the left hand side. A partial right scapula is the only preserved element of the pectoral girdle. The right forelimb is missing the proximal humerus and manus, and the left forelimb is disarticulated, lacking the manus. Some fragmentary pelvic elements remain, including both ilia. The left hindlimb is articulated but damaged, missing part of the femoral midshaft, the proximal tibia and fibula, and distal tarsals. The right hindlimb is articulated but missing both the proximal femur and the distal phalanx on digit I. The counterpart preserves two osteoderms and fragments of skull material embedded within the impressions. There is some dendritic mineral growth propagating from the skeleton.

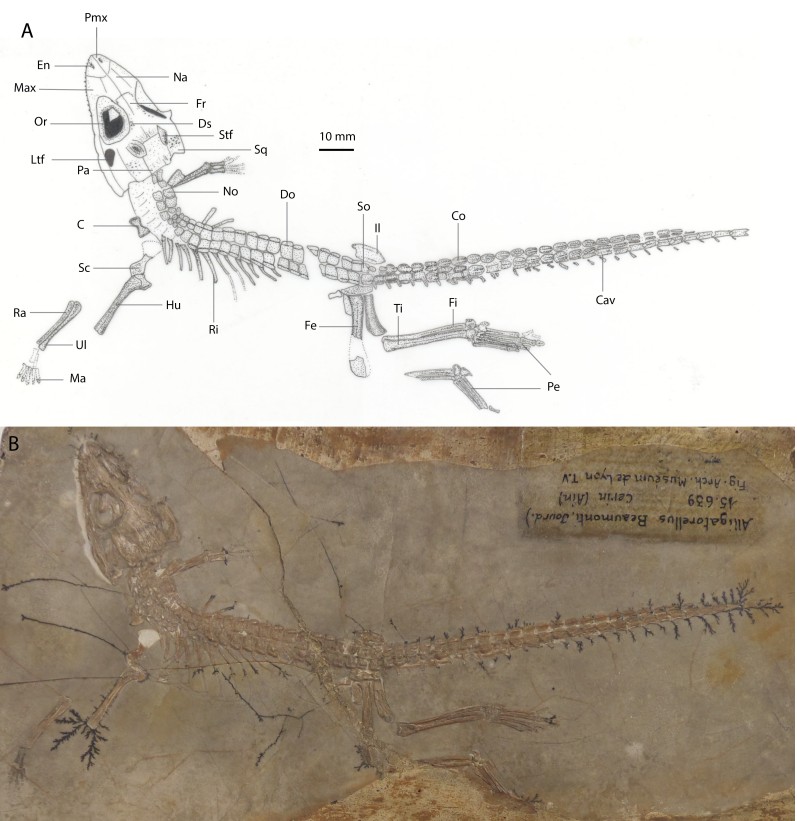

**Figure 3** (A) Line drawing of holotype specimen of *Alligatorellus beaumonti* (MNHN 15639) in dorso-lateral view; (B) photograph of holotype specimen.

*Preservation of referred specimen:* The entire skeleton is laterally flattened on a brick-red and grey slab of lithographic limestone. No counterpart is preserved. The skull is ventrolaterally flattened, exposing only the ventral and sinistral sides of the mandible, the ventrolateral portion of the skull, and nine maxillary teeth. The right forelimb is preserved only as an impression, as are the posterior-most caudal vertebrae. Otherwise, the entire axial skeleton is preserved, together with three ribs (and several rib impressions), and the left pectoral and pelvic girdles. Both hindlimbs are complete. A single row of osteoderms is preserved along the nuchal-dorsal series. The cervical vertebrae are recurved slightly posteriorly, and the posteroventrally deflected limbs give the impression of hanging loosely from the trunk.

*Additional comments: Wellnhofer (1971)* provided a detailed description of both specimens of *Alligatorellus beaumonti*. Here, we provide only a revised diagnosis as the basis for its taxonomic discrimination from the Bavarian specimens of *Alligatorellus*. Using linear morphometrics, *Wellnhofer (1971)* regarded the Cerin and Bavarian specimens to be of similar, adult ages, and largely based his justification for recognising two distinct taxa on the relatively smaller size of the Cerin specimens (which are approximately 50 mm shorter in total length). However, size and geographical distribution are not the only attributes demarcating the two as distinct taxa, as outlined below.

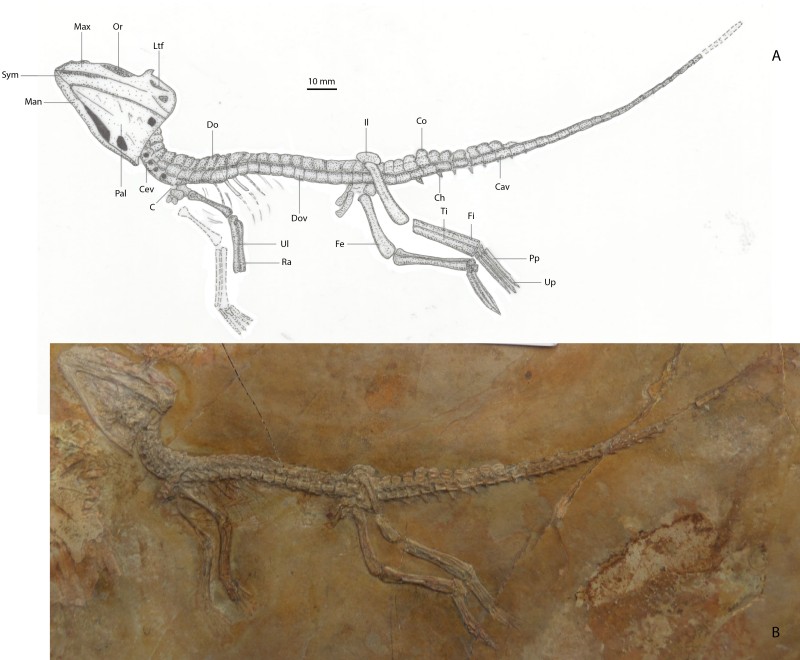

**Figure 4** (A) Line drawing of referred specimen of *Alligatorellus beaumonti* (MNHN 15638) in dorsoventral view; (B) photograph of referred specimen.

*Revised diagnosis*: *Alligatorellus beaumonti* can be diagnosed based on the following unique combination of characters and autapomorphies (highlighted with an asterisk): (1) smooth contact between maxilla and jugal (Fig. 6); (2*) frontal with unsculpted posterior and anterior portions; (3) surface of rostrum notably less sculpted than cranial table; (4) relatively large lateral temporal fenestra, approximately 30% the size of the orbit; (5*) medial longitudinal depression on posterior portion of nasal and anterior portion of frontal; (6*) frontal width between orbits narrower than nasals; (7) smooth and unsculpted region on anterior portion of squamosal nearing orbit and posterolateral process of squamosal; (8*) vertebral centra shape grades continuously posteriorly from cylindrical to elongate-spool; (9) secondary osteoderms in caudal series present; (12*) lateral ridge on sacral osteoderms forms an incipient posterior projection; (10) ratio of femur to tibia high (1.11).

*Alligatorellus bavaricus Wellnhofer, 1971*
*Alligatorellus beaumonti bavaricus Wellnhofer, 1971*

*Note on taxonomy*: *Wellnhofer (1971)* regarded LMU 1937 I 26 as the holotype of *A. beaumonti bavaricus*, and we elect this specimen as the holotype of *A. bavaricus*, which we re-rank from subspecies to species level.

*Holotype specimen*: LMU 1937 I 26 (Fig. 5).

*Referred specimen*: *Wellnhofer (1971)* also described a second specimen of *A. bavaricus*, held in the private collection of E. Schöpfel. Based on the images and description provided by *Wellnhofer (1971)*, we follow this referral. However, in view of the fact that this specimen
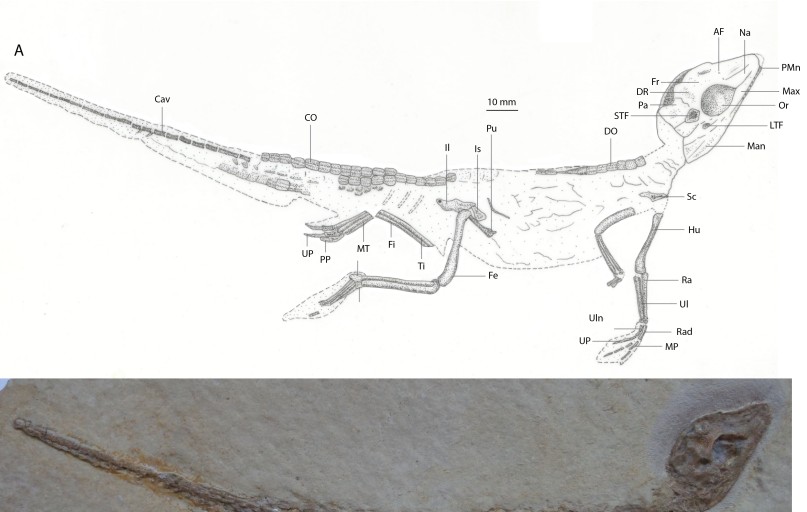

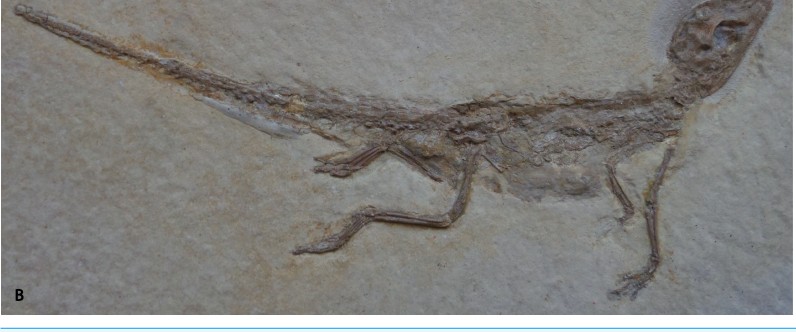

**Figure 5** (A) Line drawing of holotype specimen of *Alligatorellus bavaricus* (LMU 1937 I 26) in dorsolateral view; (B) photograph of holotype specimen.

remains in a private collection and is not publicly accessible, this referral is informal and is used only to draw attention to the existence of a second specimen.

*Type locality and horizon*: Solnhofen beds near Eichstätt, southeast Germany; early Tithonian (Late Jurassic, *Hybonoticeras hybonotum* zone; *Wellnhofer, 1971*).

*Preservation:* The specimen is a semi-three-dimensional body fossil preserved obliquely on a slab of Solnhofen 'Plattenkalk', and is fully articulated with its head dorsally recurved. As preserved, the spinal column is rod-like with a slight ventral flex, and the limbs are splayed out beneath the trunk. Trunk elements (posterior cervical and dorsal vertebrae, ribs, and osteoderms) are mostly damaged and crushed beyond recognition in an agglomeration, where there is a noticeable trace of soft tissue residue. Poor skeletal preservation means that the anterior-most vertebrae (atlas, axis, and anterior cervical vertebrae) are indistinguishable from one another. Only the eleven anterior-most dorsal paravertebral osteoderms are substantially preserved with a minor and variable degree of caudal imbrication. The next four osteoderms in the series are missing (anteriorly adjacent to the sacrum), but twenty five paired osteoderms are preserved along the tail. Poorly preserved ventral osteoderms are part of the agglomeration around the torso, and are present along the sacrum and tail. The ventral osteoderms terminate posteriorly at the same position as the dorsal series.

*Etymology of species name: bavaricus*, based on the area of the type locality, and also the sub-species name provided by *Wellnhofer (1971)* for this specimen.

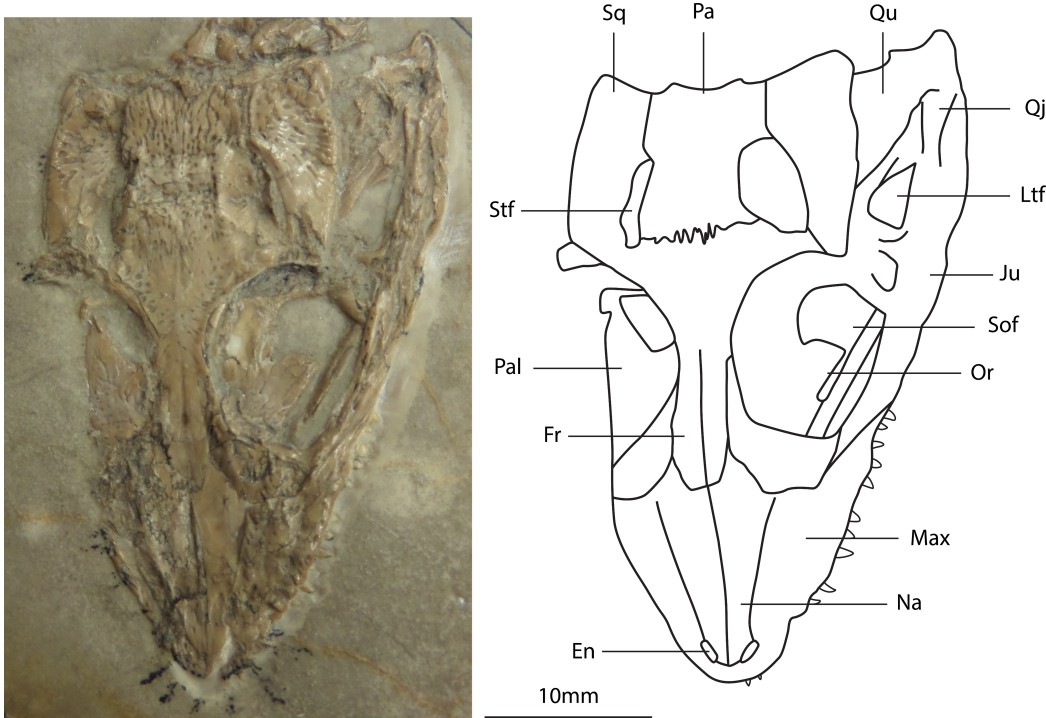

**Figure 6** Photograph and line drawing of the skull of the holotype specimen of *Alligatorellus beaumonti* (MNHN 15639) in dorsal aspect.

*Additional comments*: The majority of the features *Wellnhofer (1971)* proposed in the original diagnosis of *A. bavaricus* characterise atoposaurids in general, or are more widespread within Atoposauridae. For example, an 'acute-triangular skull with large orbit' is a general feature seen in many crocodyliforms, including all known atoposaurids and bernissartiids, and the 'biserial osteoderm shield from the nuchal to caudal region' is found in the atoposaurids *Theriosuchus* (*Owen, 1879*) and *Alligatorium* (*Wellnhofer, 1971*), and may be synapomorphic for Atoposauridae.

*Diagnosis: Alligatorellus bavaricus* can be diagnosed based on the following unique combination of characters and autapomorphies (highlighted with an asterisk): (1*) extremely narrow and short skull (ratio of skull width to orbit length is 1.29; Fig. 7); (2*) posterior surface of nares longitudinally crenulated; (3) small, slit-shaped antorbital fenestra, enclosed by nasals; (4*) prominent transverse ridge defining frontal–parietal suture, medial to supratemporal fenestrae; (5) smooth posterior region of parietal dorsal surface; (6*) dorsal osteoderms with longitudinal medial ridge, becoming more laterally placed anteriorly; (7) isometric caudal osteoderm morphology; (8*) distinct ridge on proximodorsal edge of scapula; (9*) an extremely high humerus to ulna ratio of 1.45; (10*) an extremely low femur to tibia ratio of 1.04; (11*) an extremely low tibia to ulna ratio of 0.64; (12) metatarsals I–IV equidimensional.

*Differential diagnosis to A. beaumonti*: *Alligatorellus bavaricus* can be distinguished from *A. beaumonti* based on possessing the following features: (1) proportionally larger

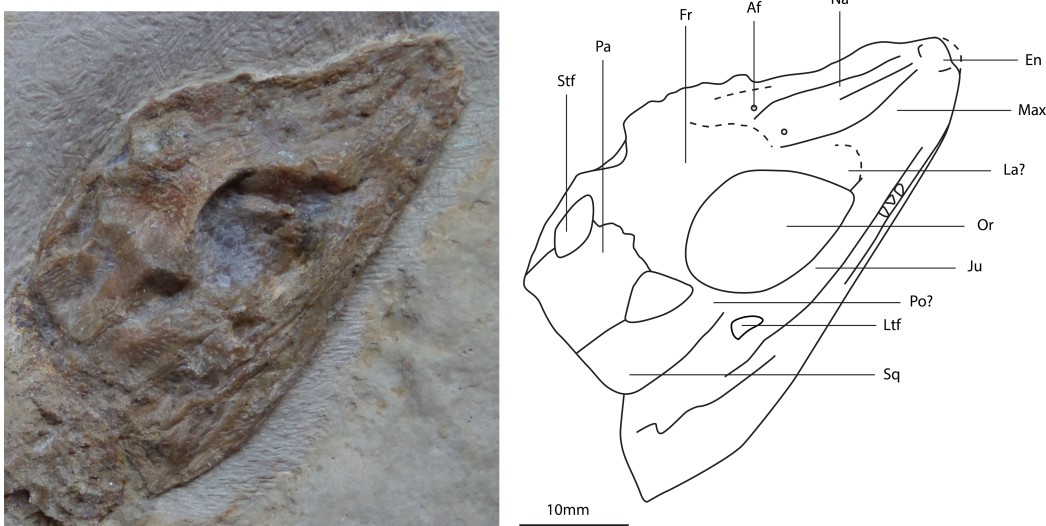

**Figure 7** Photograph and line drawing of the skull of the holotype specimen of *Alligatorellus bavaricus* (LMU 1937 I 26) in dorsolateral aspect.

orbits; (2) longitudinal crenulations on the posterior external surface of the nares; (3) a diminutive antorbital fenestra; (4) frontals proportionally wider between orbits than nasals; (5) prominent transverse ridge defining the frontal–parietal suture on the cranial table; (6) lack of posterolateral squamosal process; (7) medially-placed dorsal keels on dorsal osteoderms; (8) osteoderm shapes are isometric down length of body; (9) humerus proportionally longer than ulna (1.45 to 1.12); (10) higher ratio of humerus to femur length (0.89 to 0.75).

The electronic version of this article in Portable Document Format (PDF) will represent a published work according to the International Commission on Zoological Nomenclature (ICZN), and hence the new names contained in the electronic version are effectively published under that Code from the electronic edition alone. This published work and the nomenclatural acts it contains have been registered in ZooBank, the online registration system for the ICZN. The ZooBank LSIDs (Life Science Identifiers) can be resolved and the associated information viewed through any standard web browser by appending the LSID to the prefix "http://zoobank.org/". The LSID for this publication is: urn:lsid:zoobank.org:pub:B7CC4367-4203-4AED-8C30-2D7E4E71665D. The online version of this work is archived and available from the following digital repositories: PeerJ, PubMed Central and CLOCKSS.

## Description and comparisons of *Alligatorellus bavaricus*

The following description is solely of the type specimen LMU 1937 I 26 but, based on the images presented in *Wellnhofer (1971)*, the referred specimen does not appear to differ in any notable way. Elements of the skull of the type are fully fused, and vertebrae display complete neurocentral fusion, implying that this specimen of *Alligatorellus* had reached a mature stage of growth (*Joffe, 1967*). Measurements are provided in Table S1.

*Skull*: Observations of the skull are restricted to the dorsal and right-lateral surfaces. These external surfaces display a moderate degree of sculpting, although to a lesser extent than that of *Theriosuchus* (*Owen, 1879*; *Brinkmann, 1992*; *Wu, Sues & Brinkman, 1996*; *Schwarz & Salisbury, 2005*) and *Alligatorium* (*Wellnhofer, 1971*). The skull has an acute-triangular morphology (platyrostral) in dorsal view, typical of atoposaurids, with concave lateral margins along the relatively short snout. The intramandibular angle (defined as the angle between the lateral extremities of the cranial table and the distal snout tip, in dorsal aspect) is slightly greater (37°) than that of *Theriosuchus* (30–32°). Several teeth are preserved *in situ*, and are peg-like (pseudocaniniform), pointed and possess apicobasally and mesiodistally oriented, parallel striations. None of the teeth appear to be serrated, and in general aspect they are indistinguishable from the teeth observed in the Cerin specimens of *Alligatorellus beaumonti*. *Alligatorellus bavaricus* may possess one more maxillary tooth than the French species, although this is difficult to confidently assess due to the mode of preservation. The dentition of *Theriosuchus* (*Owen, 1879*; *Joffe, 1967*; *Brinkmann, 1992*; *Martin, Rabi & Csiki, 2010*) is substantially different in that it is heterodont. No palatal elements are visible, and aspects of the anatomy of the premaxilla, maxilla, nasals and external nares are difficult to discern due to dorsal flattening into the matrix and mandible, and because of the absence of the distal snout tip. The occipital region of the skull is also obscured by matrix and crushed, granular bone fragments, which probably represent the anterior-most elements of the axial skeleton.

There is a ventrolateral notch between the premaxilla and maxilla but, unlike in *Theriosuchus ibericus* (*Brinkmann, 1992*) and *Theriosuchus sympiestodon* (*Martin, Rabi & Csiki, 2010*), this is not occupied by an enlarged tooth. The paired nasals contribute to the external nares via a sagittal anterior projection, as in *Alligatorellus beaumonti*, *Alligatorium meyeri*, and *Theriosuchus pusillus*. *Wellnhofer (1971)* regarded this feature as diagnostic of *Alligatorellus*. However, it may be a synapomorphy of all atoposaurids: in other crocodyliforms with divided external nares, this division is formed by a sagittal projection of the premaxillae, e.g., the metriorhynchid *Maledictosuchus* (*Parilla-Bel et al., 2013*), whereas the external nares are fully open or only partially divided posteriorly in eusuchians (e.g., *Delfino et al., 2008*). A pair of small, slit-like antorbital fenestrae are present and are entirely enclosed by the nasals, a feature absent in *A. beaumonti*, but present within all specimens of *Theriosuchus* for which the snout is preserved; as such we consider this feature to be locally diagnostic of *A. bavaricus* within non-*Theriosuchus* atoposaurids. The dorsal surface of the nasals is sculpted by faint longitudinal crenulations, a feature unique within Atoposauridae, but also present in the goniopholidid *Eutretauranosuchus delfsi* (*Smith et al., 2010*; *Pritchard et al., 2013*). As such, this feature is considered a local autapomorphy of *A. bavaricus*. Posterior to the external nares, the lateral margins of the nasals are straight, contrasting with the concave margins observed in *A. beaumonti*. The dorsolaterally facing orbits are large with respect to the cranium, occupying about one third of the total cranial length and the majority of the skull width. This is comparable to *Atoposaurus oberndorferi* but distinct from *A. beaumonti*, in which the orbits occupy one quarter of the skull length. The relatively large size of the orbits might represent retention

of a paedomorphic characteristic (*Joffe, 1967*). A large amount of secondary calcite growth is present within the orbit, obscuring much of the internal cranial morphology. The right lateral temporal fenestra is deep and arcuate in cross-sectional morphology, but largely obscured as a result of the crushing of the skull. It is separated from the orbit by a mediolaterally-oriented postorbital bar, which descends steeply into the posterolateral internal margin of the orbit. The lateral temporal fenestra is similar in size to the dorsally located supratemporal fenestra, and is approximately a quarter of the size of the external opening of the orbit.

The frontals are mediolaterally concave, to a slightly greater degree than the parietals, and become extremely thin at the orbital margin, lacking the elevated orbital rims seen in *Theriosuchus* (*Owen, 1879*). Compared to the nasals, they are relatively wide with respect to the frontals in *A. beaumonti*. The anterior frontal ramus extends slightly beyond the anterior tip of the prefrontal, a feature which we consider to be a local autapomorphy because of its absence in other atoposaurids, but that is present in some other non-eusuchian neosuchians, including *Eutretauranosuchus delfsi* (*Pritchard et al., 2013*) and *Pholidosaurus purbeckensis* (*Salisbury, 2002*; *Montefeltro et al., 2013*). The anterior contacts between the frontals, prefrontals and lacrimals are largely obscured, as is the overall morphology of these pre-orbital elements. However, the majority of the anterior margin of the orbits comprises a deep and thick wedge of bone that descends as a vertical sheet into the orbit, forming a distinctive anterodorsal brow. The maxilla contributes extensively to the ventral margin of the orbit, with the contact between the maxilla and the lacrimal becoming indiscernible more anteriorly as a result of the mode of preservation. The jugal occupies half of the ventral margin of the orbit, posterior to the maxilla. Palpebrals were either absent or are not preserved, but appear to be present in the anterior orbit of *Alligatorellus beaumonti*.

Posterior to the orbits, the dorsal surface of the skull is mildly sculpted by anisotropic and heterogeneously spaced pits that are similar to *Alligatorellus beaumonti*, but are less prominent than those seen in *Theriosuchus* and *Alligatorium*. In contrast, this surface is smooth and unsculpted in *Atoposaurus* (*Wellnhofer, 1971*; J Tennant, pers. obs., 2013). It is plausible that the heterogeneous degree of cranial sculpting seen in atoposaurids including *Alligatorellus* and *Montsecosuchus* is useful in distinguishing specimens at the species level. Between the supratemporal fenestrae is a prominent mediolateral ridge defining the suture between the frontal and parietal, a feature we consider diagnostic of *A. bavaricus*. The anterior parietal is not sculpted where it contacts the frontals, unlike *A. beaumonti* where the whole cranial table (excluding the frontals) is homogeneously sculpted with small circular pits. The squamosal is homogeneously sculpted, as with the parietal, with a dorsally convex dorsal surface and orthogonal lateral and posterior margins, differing from *Theriosuchus pusillus* which has a smooth posterolateral process (*Owen, 1879*; J Tennant, pers. obs., 2013). The cranial table is mostly flat, as is the case in most other atoposaurids, with the exception of the slightly domed structure that characterises *Montsecosuchus* (*Buscalioni & Sanz, 1990a*), and possibly *Atoposaurus*. The anterolateral portion of the squamosal is sharply pointed and curves posteromedially around the supratemporal fenestra. Here, it is

initially gently arcuate, then becomes straight as it contacts the parasagittally-directed and straight medial edge. This gives the squamosal an overall distorted rhombohedral shape in dorsal aspect. The majority of the dorsomedial margin of the squamosal contributes to the supratemporal fenestra, with the lateral portion obscuring most of the ventrally-placed quadrate and quadratojugal. The posterolateral process of the squamosal is greatly reduced compared to other atoposaurids, in which it generally tapers to a point, and is therefore considered to be a local autapomorphy of *A. bavaricus*, being similarly present in other basal neosuchians such as *Amphicotylus lucasii* (*Mook, 1942*). In *Alligatorellus beaumonti*, there is no development of the posterolateral process, the posterior edge instead being slightly anterolaterally directed. Between the supratemporal fenestrae, the paired, rectangular parietals are as mediolaterally wide as the frontals between the orbits. The parietals contribute to the posteromedial margin of the supratemporal fenestra, but the relationship with the postorbitals is difficult to see due to post-mortem damage. However, the postorbital bar is present and weakly developed, possessing a superficial furrow connecting the orbit and the supratemporal fenestra. The frontal only contributes to the supratemporal fenestra at its anteromedial edge. Here, the frontal and parietal form a lateral wedge, which thins laterally into the postorbital bar. The posterior portion of the dorsal surface of the parietal is smooth, a feature otherwise only found in *Atoposaurus*, although in that taxon the skull is entirely unsculpted (*Wellnhofer, 1971*; J Tennant, pers. obs., 2013); as such, we consider this heterogeneous pattern of cranial sculpting to be autapomorphic for *A. bavaricus*. The lateral and ventral surfaces of the skull are largely obscured by the displaced and crushed mandible, and the preserved orientation of the skeleton.

The mandible is not visible ventral or anterior to the orbit, and is largely obscured posteriorly. It has been slightly dorsally displaced into the ventrolateral portion of the right-lateral face of the skull. The mandible broadens posteriorly both mediolaterally and dorsoventrally, developing a lateral shelf as it flares out beneath the lateral temporal fenestra, possibly at the position at which the mandibular fenestra would have been situated. The ventral margin of the mandible curves medially and substantially thins mediolaterally at its posterior extremity, where it forms an acute and recurved process, the posterior margin of which is gently concave and slightly set back from the posterior edge of the cranial table.

*Axial skeleton:* One of the most striking features of atoposaurids is that the tail length is greater than the length of the torso, and comprises approximately one-half of the total length of the skeleton. In *Alligatorellus bavaricus* there are seven cervical (including the axis and atlas) and fifteen dorsal vertebrae (note that *Wellnhofer (1971)* observed only seventeen presacral vertebrae, using osteoderm count as a proxy). These vertebrae are mostly indistinguishable from one another, but their presence is estimated based on their associated dorsal paravertebral osteoderms which, along with the poor preservation of the trunk region, largely obscure the morphology of the vertebral column. As noted by *Wellnhofer (1971)*, three sacral vertebrae seem to be present, but their preservation means that this cannot be determined with any certainty, with all elements crushed beyond distinction. If correctly determined, sacral count might be a distinguishing feature

between *A. bavaricus* and *A. beaumonti*, with the latter only having two sacral vertebrae, but variation in sacral count is difficult to discern in atoposaurids due to poor preservation of the axial skeleton in specimens of *Alligatorellus*. There are around forty caudal vertebrae, although the precise number is difficult to determine, with the distal-most two or three absent, as indicated by impressions. Much of the caudal vertebral series is variably covered in matrix and fixing glue, obscuring most of the morphological detail and intervertebral articulations. In the central caudal series, a melange composed of dorsal and ventral paravertebral osteoderms obscures much of the anatomical detail. Only the first four caudal vertebrae can be used to observe any of the anatomy from a right-lateral perspective. It is unknown whether the vertebrae were procoelous, as in *Theriosuchus* and eusuchians (e.g., *Pol & Gasparini, 2009*), or amphicoelous.

The dorsal osteoderms occur in a biserial row from the anterior-most cervical vertebrae to about the mid-point of the caudal series, a feature that characterises all unambiguous atoposaurids, with the exception of *Atoposaurus*, and that is also absent in the putative atoposaurid *Karatausuchus* (*Efimov, 1976*; *Storrs & Efimov, 2000*). The osteoderms of *A. bavaricus* are imbricated along their entire length, and there is no 'peg and socket' articulation as described in two scutes assigned to *Theriosuchus pusillus* (*Owen, 1879*; *Schwarz-Wings et al., 2011*) and in *Theriosuchus guimarotae* (*Schwarz & Salisbury, 2005*). The osteoderms of *A. bavaricus* are rounded, and the lateral edges are predominantly convex, with one or two being marginally concave. There is a central longitudinal ridge on the dorsal surface of osteoderms of *A. bavaricus*, similar to some of the caudal osteoderms in *Theriosuchus*, but contrasting with *Alligatorium meyeri* and other atoposaurids. The degree of sculpting on the osteoderm dorsal surfaces increases posteriorly, as does the prominence of the longitudinal keel which shifts to a slightly medial position from an initially more central position, unlike *Alligatorellus beaumonti* in which it is consistently laterally placed as a distinct shelf. The lateral and medial edges of the osteoderms are smooth and either straight or convex, and the straight anterior and posterior margins are parallel. The morphology of the ventral osteoderm series is very similar, where visible, but with more prominent longitudinal ridges in the more posterior elements. There is no visible morphological heterogeneity in the nuchal and sacral osteoderms, contrasting with *Alligatorellus beaumonti* in which this feature is highly distinctive. It is unknown whether the ventral series are paired or not in *A. bavaricus*, as the ventral portion of the skeleton is mostly unobservable.

Other minor axial elements are partially visible beside the osteoderms. Two thoracic ribs are preserved embedded within the trunk melange. They are gently arcuate in their overall morphology, and not preserved *in situ*. There are several other rib elements more anterior to these and just ventral to the anterior-most osteoderms, but they are largely obscured by the overlying matrix and axial elements. Three posteroventrally directed chevrons are *in situ* with their proximal caudal vertebrae, positioned just posterior to the only visible three-dimensionally preserved vertebrae.

*Pectoral girdle*: Only the right scapula is preserved, and is fragmented at both ends, including both the glenoid fossa and coracoidal contact. It is bow shaped, with a distinct

dorsoventral contraction and mediolateral thickening into a compressed cylindroid at mid-length. The dorsal surface becomes thin and sharp anteriorly, culminating in a broad and deep, basin-like medial depression, contrasting with *Montsecosuchus depereti* in which the entire element is flat (*Buscalioni & Sanz, 1990a*). The proximodorsal edge overhangs this depression, a feature not observed in other atoposaurids, and is considered to be a diagnostic feature of *A. bavaricus*. Posteriorly, the scapula flares out in a similar fashion to the anterior blade, but the distal portion is mostly absent, so the complete morphology is unknown. A posteroventral process projects out from the posterior blade, twisting from the ventral surface into a short, thickened rod.

*Forelimbs*: The right forelimb is nearly complete, with an articulated humerus, radius and ulna, but the manus is crushed. The proximal third of the humerus is also crushed, with the external cortices of the exposed shaft removed, revealing the internal bone. The humerus expands slightly proximally, and the shaft is straight and broader mediolaterally than anteroposteriorly. The morphology of the deltopectoral crest cannot be determined. The radial condyle is broad and directed anteriorly. The distal articular surface of the humerus is strongly rugose, and oriented at 40° to the long axis of the shaft. The shaft is relatively straight, similar to more advanced neosuchians such as *Shamosuchus* (*Pol, Turner & Norell, 2009*). The anterior intercondylar groove is not visible, but the supracondylar fossa forms a deep posterior furrow, terminating a short distance up the shaft, and is bound medially by the relatively weaker ulnar condyle, the morphology of which is mostly obscured. The external surfaces of the condylar heads are smooth. The humerus is slightly shorter than that of *A. beaumonti*, but the radius is proportionally longer. The stylopod to zeugopod ratio in both limbs is proportionally lower than in all other atoposaurids, a feature that we consider diagnostic of *A. bavaricus*.

The radius is slightly longer than the more robust ulna, the two resting against each other without twisting sharply; as such the respective proximal and distal articular surfaces have long axes in the same orientation. The radius is gently longitudinally arcuate in its proximal third, conforming to the gentle curvature of the distal ulnar shaft. The radial head is mediolaterally expanded, and is about two-thirds the size of the ulnar head it rests against. The ulnar head is damaged, and the radial head and the associated humeral condyle actually appear quite mismatched in size, suggesting a large volume of cartilage or muscle attachment at this joint, also emphasised by the heavily rugose articular surface. The lateral part of the radial shaft thins to about 70% of its width and becomes ridge-like at around two-thirds of its length. The ulnar shaft is equidimensional through its entire length, and finishes with a triangular-shaped distal articular surface. The carpus cannot be fully observed.

Little of the left forelimb is preserved: the distal humerus is crushed, with the proximal ulna and entire radius missing, preserved only as impressions. However, aspects of the morphology of the carpus can be observed. The radiale is long and slender, with expanded proximal and distal ends, much like *A. beaumonti* in which the elements are well-preserved in the holotype. The ulnare is slightly shorter, with a stronger mediolateral compression of the shaft, and overall more gracile morphology. In *A. beaumonti*, the ulnare has a proximal

groove on the lateral surface, terminating at 80% of the length of the element, but whether this is present in *A. bavaricus* cannot be determined. However, the ulnare in *A. bavaricus* is not 'hatchet shaped' as in *A. beaumonti* or the specimen assigned to *Alligatorellus* sp. by *Schwarz-Wings et al. (2011)*. Furthermore, the radiale in *A. beaumonti* is larger than the ulnare, distinguishing the two species of *Alligatorellus*. All additional carpal elements in *A. bavaricus* are crushed to the point where their morphology cannot be meaningfully observed. The entire manus is bent backwards, indicated by its impression and in a similar manner to the pedal orientations. All of the elements are highly distorted and crushed, with only moderate lateral compression indicated by the slight crushing of the more gracile elements.

*Pelvic girdle*: Only fragments of the pelvic girdle are preserved. The ilium forms an elongated S-shape in dorsal view, and is thickened anteriorly. Much of the morphology is obscured by the orientation of the specimen on the rock slab, but the postacetabular process appears to be fenestrated at its tip (although this might be a post-mortem artefact), greatly thickened, and leads into a deep and broad acetabulum. An element just below this on the slab is one of the pubes. Much of the morphology is again obscured by the orientation in which it is embedded in the matrix. The proximal head is expanded into a broad wedge-shape and twists slightly to become oblique to the stouter distal end, which is more circular in cross section. The proximal portion of the shaft is transversely flattened and sub-elliptical in cross-section, and has a strongly rugose surface, partially obscured by an overlying displaced rib. There is a fan-shaped structure situated anterior to the ilium, which we interpret as a fragment of the anteriorly displaced ischium. The distal end is thin and gently convex, with a slightly crenulated distal extremity. Gentle striations from the distal end are directed towards the transversely thickened shaft, which increases in breadth more proximally on the dorsal margin and has a more slender ventral margin. The proximal end is hidden underneath the skeleton so that the remaining morphology cannot be observed.

*Hindlimbs*: Overall, the hindlimbs are about 1.4 times the length of the forelimbs. The right hindlimb is mostly complete with a laterally flattened tarsus and pes. The femur is missing from the left hindlimb (although it is possibly hidden underneath the skeleton), and the tibia and fibula are both crushed. The left pes is well-preserved, with partially crushed tarsal and pedal elements. The femur is the most robust limb bone of the skeleton, and is morphologically similar to the ulna, being gently sigmoidal down the length of the shaft. The femoral head is moderately expanded and equidimensional to the distal end of the femur. The femoral head grades smoothly into the posteriorly placed fourth trochanter, which is weakly developed, ridge-like, and distally thickened, terminating at one-sixth of the length from the proximal end. Adjacent to this, on the lateral surface, there is an accompanying groove for attachment of the femoral-pelvic musculature. The distal end of the right femur is damaged and fractured, and the distal condylar morphology cannot therefore be determined.

The left tibia and fibula are mostly concealed within the slab and underneath other bones, and only the straight shafts are exposed. The lateral surfaces of both elements from

the right hindlimb are fully exposed, and demonstrate that they are equal in length to the femur. Both ends of the tibial shaft are anteroposteriorly compressed, with the distal end slightly more so. The proximal portion of the tibia is slightly posteriorly deflected, but to a lesser degree than in *Alligatorellus beaumonti*. The tibial shaft becomes slightly anteroposteriorly expanded at mid-length. Distally, the lateral margin of the tibia thins anteroposteriorly, culminating in a sharp ridge at the distal end, and resulting in a triangular cross-section. The proximal half of the fibula is gently twisted to accommodate the mid-tibial expansion, and articulates with the posterior face of the proximal head of the tibia. As a result of the fully articulated nature of the tibia and fibula, the morphology of the proximal and distal articular surfaces is obscured. Furthermore, the distal end of the fibula is damaged. In lateral view, the fibula is much more slender than the tibia, and has a more circular cross section than the elliptical to triangular tibia. The astragalus is not visible in either hindlimb. The calcaneum is present, but is obscured by matrix and glue.

On the right hindlimb, metatarsals I–III and part of metatarsal IV are preserved, as well as a poorly preserved, vestigial fifth metatarsal that is less than one-third the length of the other four metatarsals. Their long axes are parallel to one another, with the proximal and distal ends resting against each other. The nature of the distal articulations is obscured. The left pes is preserved in an oblique view, and provides a better perspective of the metatarsal morphology, although metatarsal V is not visible. The tarsal phalangeal formula, as stated by *Wellnhofer (1971)*, is 2-3-4-4-(1). The metatarsals are long, gracile, and transversely expanded at their proximal ends with an overall similar morphology to one another. Their distal ends have been slightly anteroposteriorly compressed, and the straight shafts all have an elliptical cross-section. On the left pes, the proximal tip of metatarsal I is obscured beneath metatarsals II–IV but, where visible, the metatarsal is anteroposteriorly compressed, and twists anteromedially towards its distal end, at which point it thickens and broadens into a sub-oval cross section. The distal articular surface of metatarsal I is only partially visible; this rugose surface curves medially to occupy the distal-most edge of the medial surface. Metatarsal II is slightly longer than metatarsal I, with a mediolaterally compressed proximal end, and a ventral surface that forms a thin ridge. Metatarsal II gradually thickens distally, and the shaft twists in a similar manner to metatarsal I, but instead the ventromedial edge becomes more prominent as a ridge, bounding the medial edge of a small distal depression on the ventral surface. The distal end of metatarsal II is convex, and the articular surface is obscured. Most of metatarsal III, except for the shaft, is obscured, with the shaft appearing to be as long as metatarsal II but thickened to a lesser degree distally. Metatarsal III is slightly more gracile than the others. The sharpness of the proximoventral ridge is also less apparent in metatarsal III. Metatarsal IV is mostly obscured, but has a straighter, less twisted shaft that is more continuously oval in cross-sectional morphology than the metatarsals.

## Additional material previously referred to *Alligatorellus*

Atoposauridae indet.
*Alligatorellus* sp. *Schwarz-Wings et al., 2011*

*Specimen:* MfN MB. R. 4317.1-12, a partial disarticulated skeleton.
*Locality and horizon:* Kelheim, Bavaria, Germany; early Tithonian, *Hybonotum* Zone, *Rueppelianus* Subzone (*Schwarz-Wings et al., 2011*).
*Preservation:* Disarticulated axial and appendicular elements adjacent to a single row of paravertebral osteoderms. Some limb elements have been prepared out of the matrix.

   *Comments:* An additional specimen from Bavaria was recently assigned to *Alligatorellus* sp. by *Schwarz-Wings et al. (2011)*. This is a substantially larger individual than the four known specimens comprising *A. beaumonti* and *A. bavaricus*, and is represented by a disarticulated, partial postcranial skeleton. With the revised diagnosis presented above for *Alligatorellus*, the only comparable diagnostic material is the osteoderms, which differ in morphology to those of *A. beaumonti* and *A. bavaricus*. Distinguishing features present in MfN MB. R. 4317.1-12 include: (1) dorsal osteoderms are square-shaped, rather than rectangular, with a possible anterior articular process (similar to the 'peg and socket' morphology seen in some specimens of *Theriosuchus* and goniopholidids); (2) dorsal osteoderms are distinctly asymmetrical about their long-axis; (3) ventral osteoderms bear a series of nutrient foramina, as well as an anteroposteriorly oriented ridge along their anterior portions (although note that this aspect of the osteoderms is not visible in any specimen of *Alligatorellus beaumonti* or *A. bavaricus*); and (4) caudal osteoderms are often laterally serrated, grading from a narrow to elongated elliptical shape. Several of these features regarding osteoderm morphology may be diagnostic within Atoposauridae. An additional difference is the more laterally than medially expanded proximal end of the radiale, with a proximodistally oriented crest extending along the anterior surface of the shaft. This, together with the outlined differences in osteoderm morphology indicates, that MfN MB. R. 4317.1-12 may represent a distinct atoposaurid taxon, or another species of neosuchian outside of Atoposauridae.

   Based on the revised diagnosis for *Alligatorellus* presented in this study, and the notable differences in preserved osteoderm morphology, it is questionable whether the specimen described by *Schwarz-Wings et al. (2011)* can be assigned to *Alligatorellus*. Its initial referral to this genus was based on several lines of evidence, including the longitudinally elliptical shape of the caudal osteoderms, a feature otherwise only seen in the distal-most caudal osteoderms of *Alligatorellus beaumonti*, although they are more rectangular in the French taxon. However, the morphology of the osteoderms of MfN MB. R. 4317.1-12 is similar to the dorsal osteoderms of *Montsecosuchus depereti*, including the presence of a continuous, medially-positioned keel along the external surface (*Buscalioni & Sanz, 1990a*; J Tennant, pers. obs., 2013), but *Montsecosuchus* does not preserve any osteoderms of similar size or morphology to the imbricated series preserved in MfN MB. R. 4317.1-12. The imbrication of these dorsal osteoderms cannot be used to assign MfN

MB. R. 4317.1-12 to *Alligatorellus*, as this is a feature also present in *Theriosuchus pusillus* and *Alligatorium*, the putative atoposaurid *Brillanceausuchus*, and other neosuchians including *Pachycheilosuchus* (*Rogers, 2003*; J Tennant, pers. obs., 2014). The higher degree of sculpting of the osteoderms was regarded as ontogenetic variation by *Schwarz-Wings et al. (2011)*, but all other specimens of *Alligatorellus* also appear to represent mature individuals. Therefore, the greater degree of sculpting observed in the osteoderms of MfN MB. R. 4317.1-12 may represent a taxonomic difference. The position of the dorsal keel on these osteoderms, and the lack of symmetry in their outlines in dorsal aspect also represent differences between the osteoderms of MfN MB. R. 4317.1-12 and those observed in other specimens assigned to *Alligatorellus*. Additionally, the limb ratios presented in *Schwarz-Wings et al. (2011)*, p. 203, Table 2) imply that this specimen is allometrically quite distinct from *Alligatorellus*, and perhaps more closely related to *Alligatorium*. For now, we consider MfN MB. R. 4317.1-12 to be an indeterminate atoposaurid pending its inclusion in a comprehensive species-level phylogenetic analysis of Atoposauridae (J Tennant & PD Mannion, 2014, unpublished data).

## DISCUSSION

### Osteoderm morphology in atoposaurid systematics

The morphology of the parasagittally-arranged postcranial osteoderms of atoposaurids has not previously been regarded as an important characteristic in atoposaurid taxonomy, generally due to their relatively rare preservation *in situ* (e.g., *Buscalioni & Sanz, 1990a*; *Michard et al., 1990*; *Wu, Sues & Brinkman, 1996*). The exception to this is a study of western European specimens by *Schwarz-Wings et al. (2011)*. However, as noted here for specimens referred to *Alligatorellus*, subtle differences in osteoderm morphologies, particularly the extent, position, and continuity of the longitudinal keels on the dorsal surfaces, can prove to be diagnostic at species level.

The pattern of ornamentation on the osteoderms of atoposaurid taxa, as with other osteoderm-bearing crocodylomorphs (*Vickaryous & Hall, 2008*), is similar to that seen in dermatocranial ornamentation, particularly with respect to the dorsal surface of the skull table. Exceptions to this are *Atoposaurus* and the putative atoposaurid *Karatausuchus*, in which there is no evidence of cranial sculpting, and no evidence of preserved osteoderms (*Wellnhofer, 1971*). Furthermore, the general distribution of osteoderms in *Alligatorellus* is similar to that in basal crocodylomorphs such as protosuchians, sphenosuchians and the enigmatic taxon *Hoplosuchus kayi* (*Gilmore, 1926*; J Tennant, pers. obs., 2014), which have biserial rows of imbricated, rectangular dorsal osteoderms that might have served in a more functional support role than that proposed for atoposaurids (*Clark & Sues, 2002*; *Pol et al., 2004*). There remains the possibility that osteoderm morphology varies intraspecifically, with multiple morphotypes represented within a population, as is the case in some other archosaurs (e.g., ankylosaurs *Burns, 2008*). However, sample sizes are currently too small to ascertain if this might be the case for atoposaurids. Nevertheless, unequivocal intrageneric differences in osteoderm morphology are observed between

*Alligatorium* and *Theriosuchus* (e.g., *Owen, 1879*; *Wellnhofer, 1971*; *Wu, Sues & Brinkman, 1996*), prompting consideration of its utility for systematic placement of *Alligatorellus*.

Establishing the positional homology of osteoderms is important for evaluating taxonomic status in many tetrapod groups, including crocodylians (*Ross & Mayer, 1983*), aetosaurians (*Parker, 2007*; *Parker & Martz, 2010*), and chronosuchians (*Buchwitz et al., 2012*). This is difficult in the case of less complete or disarticulated specimens, such as that described by *Schwarz-Wings et al. (2011)* as *Alligatorellus* sp., comprising articulated and disarticulated elements which they considered to represent a single individual with heterogeneous osteoderm morphology. In *Alligatorellus*, both the positional homology and differences in morphology in the discrete axial regions are diagnostic at species level. There are four regions: cervical (or nuchal), dorsal, sacral, and caudal. These regions typically comprise continuous rows of anteroposteriorly arranged (paramedian or paravertebral) osteoderms. On the basis of osteoderm morphology and configuration, *Alligatorellus* differs from *Theriosuchus pusillus* and advanced eusuchians (e.g., *Leidyosuchus*) which have the ventral body encased within an articulating (but not overlapping or imbricating) shield of parasagittal rows of singular osteoderms (*Owen, 1879*; *Brochu, 1997*). It also differs from *Alligatorium*, in which osteoderms bear no dorsal keel, and from *Montsecosuchus* which has two to three rows of non-imbricating, and longitudinally oval dorsal osteoderms. Below, we discuss the three different morphotype series found in specimens ascribed to *Alligatorellus*.

### A. bavaricus morphotype

The dorsal keel in osteoderms of *A. bavaricus* is in a more medial position nuchally, gradually migrating laterally along the dorsal series before becoming medially placed in the sacral and caudal series (Fig. 8A). Throughout this gradation, individual osteoderms are similarly robust, but adopt an increasingly more sub-rectangular to elliptical morphology posteriorly. Whereas they imbricate in the dorsal series, this change in shape leads to them abutting one another longitudinally, with no overlap. The longitudinal keel always occupies the entire length of the dorsal surface, and becomes more prominent posteriorly. There is a caudal ventral series of secondary osteoderms, but these are few in number and do not extend beyond the anterior half of the tail. This is similar to the condition in *Montsecosuchus depereti* (*Buscalioni & Sanz, 1990a*), but contrasts with *Theriosuchus*, in which they extend to the end of the caudal series. In contrast to *A. beaumonti*, the dorsal keel observed in sacral and anterior caudal osteoderms of *A. bavaricus* never develops an incipient posterior projection. It is likely that the 'accessory osteoderms' of *Alligatorellus bavaricus* described by *Wellnhofer (1971)* are the result of incomplete osteoderm development: they appear to be mostly comprised of the longitudinal keel, which forms as part of the earliest phase of osteoderm development (*Vickaryous & Hall, 2008*).

### A. beaumonti morphotype

The biserially arranged osteoderms of *A. beaumonti* form a continuous dorsal shield, similar to *Theriosuchus pusillus* and other atoposaurids (Fig. 3). Their longitudinally imbricating arrangement is comparable to that of extant alligatoroid species such as

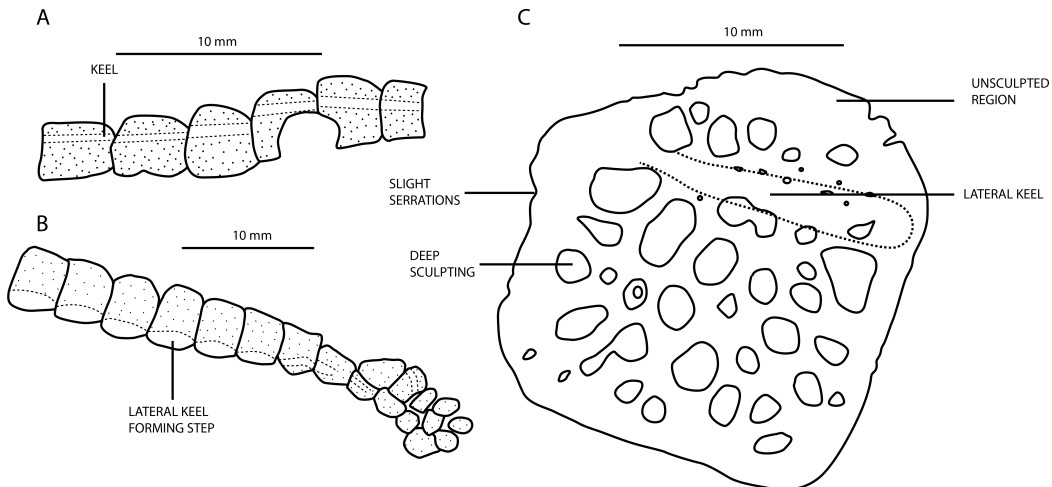

**Figure 8** (A) Line drawings of the dorsal osteoderms of *A. bavaricus*; (B) line drawings of the cervical and dorsal osteoderm series of *A. beaumonti*; (C) Line drawing of a dorsal osteoderm of the specimen described as *Alligatorellus* sp. (MfN MB. R. 4317.1-12) by *Schwarz-Wings et al. (2011)*, but considered here as Atoposauridae indet.

*Caiman crocodilus* and *Alligator mississippiensis* (*Burns, Vickaryous & Currie , 2013*), but with fewer paramedian dorsal series. The extent of the caudal ventral series is much greater than in *A. bavaricus*, forming a complete dermal coating. The distal-most osteoderms are small, seemingly under-developed, sculpted elements. In the caudal series, the longitudinal ridges are pronounced, longitudinally extensive, and medially placed, similar to *A. bavaricus*. The more sacrally placed caudal elements have less pronounced keels than *A. bavaricus*. They also become smaller and more ovate, with the ridges gradually almost disappearing, and only occupying the posterior portion of each element, whereas sculpting remains the same. This skewing of the keels is most pronounced in the dorsal and sacral osteoderms, where they form rounded protrusions on the dorsal side and become laterally displaced on the ventral series. This unusual shift is particularly evident in the dorsal series, where the lateral keel becomes more prominent and more anteroposteriorly extensive, forming a distinct step from the main body of each osteoderm (Fig. 8B). The ventral and dorsal morphology is quite similar, with the ventral keels almost seeming to diverge ventrally with each accompanying rib. The sacral and anteriormost caudal osteoderms develop an incipient lateral projection, almost appearing to diverge into two individual elements medial to this. The ventral series either terminates around the position of the third dorsal rib, or is not preserved anteriorly from this point. The dorsal series adopts a heterogeneous range of morphologies, with some elements reducing to around one-sixth the size of the other osteoderms more nuchally, and with all losing the presence of the keel. This contrasts with *Alligatorium meyeri* and *Theriosuchus pusillus*, where they are morphologically continuous.

### MfN MB. R. 4317.1-12 ('Alligatorellus sp.') morphotype

The deeper sculpting present in this specimen was ascribed to ontogenetic variation by *Schwarz-Wings et al. (2011)*, based on its larger size compared to other specimens of

*Alligatorellus* (Fig. 8C). Maturity of the type specimens of both species of *Alligatorellus* is discussed above, as are differences in osteoderm morphology, suggesting that this specimen represents a distinct taxon from *Alligatorellus*. These differences include the more medial position of the keel in MfN MB. R. 4317.1-12, and the lateral deflection of the body of the osteoderms adjacent to this. The keel is also not as longitudinally continuous in MfN MB. R. 4317.1-12 as it is in *A. beaumonti* and *A. bavaricus*. Additionally, the lateral edge is serrated, and there are unsculpted areas on the dorsal surface. Moreover, they are less robust overall than the other specimens of *Alligatorellus*, in spite of their greater size, and overall appear similar to the osteoderm ascribed to *Theriosuchus* sp. by *Wu, Sues & Brinkman (1996)*.

## The taxonomic validity of *Atoposaurus* and *Alligatorium*

*Alligatorellus beaumonti* coexisted with *Atoposaurus jourdani* and *Alligatorium meyeri* in eastern France, while *Alligatorellus bavaricus* lived alongside *Atoposaurus oberndorferi* and possibly *Alligatorium franconicum* and *Alligatorium paintenense* in southeastern Germany (*Wellnhofer, 1971*; Figs. 1, 2). This high diversity of atoposaurids in the Late Jurassic of Germany and France, combined with potential juvenile features in *Atoposaurus*, has led some to suggest that *Atoposaurus* might in fact represent a juvenile specimen of one of the other sympatric atoposaurid species (*Buscalioni & Sanz, 1988*). Furthermore, *Benton & Clark (1988)* suggested that *Atoposaurus, Alligatorellus* and *Alligatorium* might represent a single growth series.

Ontogenetic allometric variation has received considerable attention in extant crocodylians, particularly in population-level studies (e.g., *Dodson, 1975*). Through crocodylian ontogeny, several allometric relationships have been recognised in different taxa: (1) the skull lengthens, and becomes more dorsoventrally flattened and laterally compressed in *Caiman* (*Monteiro & Soares, 1997*; *Monteiro, Cavalcanti & Sommer III, 1997*); (2) the skull lengthens and widens in *Alligator sinensis* (*Wu et al., 2006*), *Crocodylus moreletii* (*Platt et al., 2009*) and *Crocodylus siamensis* (*Chentanez, Huggins & Chentanez, 1983*), as does the snout in *Alligator sinensis*; (3) reduction in relative orbit size to the skull occurs in *Crocodylus acutus, Gavialis gangeticus, Mecistops cataphractus* and *Tomistoma schlegelii* (*Piras et al., 2010*); and (4) the orbit, snout and skull shape changes through ontogeny in *Caiman latirostris* (*Verdade, 2000*). However, as *Verdade (2000)* noted, many of these allometric factors covary with both size and ontogenetic stage, and therefore it is often difficult to interpolate from these allometric relationships to determine an ontogenic stage in fossil taxa.

To test the hypothesis that *Alligatorellus, Alligatorium* and *Atoposaurus* represent a single ontogentic series, or that *Atoposaurus* is a juvenile of at least one of the other taxa, we plotted a number of anatomical measurements (skull width, snout length, and orbit length) for each of the species against skull length, and also carried out a covariance-based Principal Components Analysis (PCA) in R (*R Development Core Team, 2014*) (Fig. 9).

An increase in skull width and snout length relative to skull length is seen in both the French and German atoposaurid groups, although this is much more pronounced in

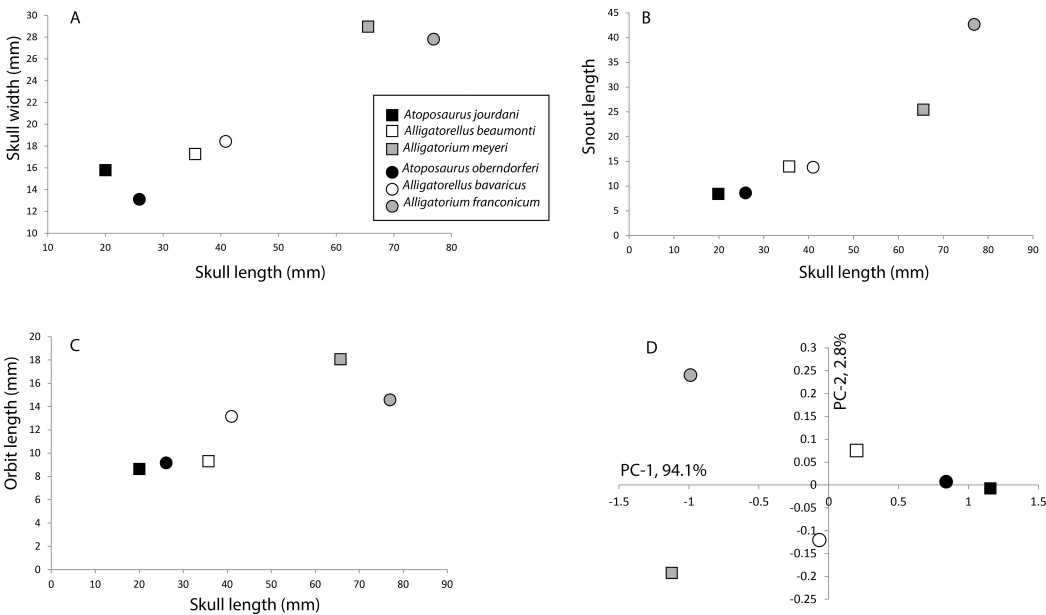

**Figure 9 Morphometric plots of the holotype specimens of the Late Jurassic atoposaurids *Alligatorellus*, *Alligatorium*, and *Atoposaurus*.** (A) Skull length versus skull width; (B) skull length versus snout length; (C) skull length versus orbit length; and (D) PCA plot for all specimens based on six primary measurements (see text and Data S1 for details). Squares represent French taxa, and circles represent German taxa.

the German specimens as a result of the longirostrine form of *Alligatorium franconicum* (Figs. 9A and 9B). However, this trend is not entirely linear, with *Alligatorellus beaumonti* having a distinctly longer, but almost equally wide, skull relative to *Atoposaurus jourdani*. A similar pattern is recorded for orbit length relative to skull length, although this trend is considerably less pronounced in the German taxa, and there is little difference between *Atoposaurus jourdani* and *Alligatorellus beaumonti*, despite an almost doubling of skull length (Fig. 9C). If *Atoposaurus*, *Alligatorellus*, and *Alligatorium* were part of the same growth series, we would expect a consistent relationship between the two geographic groups in each of these aspects, a pattern not produced here.

Our PCA of six primary measurements (skull length, skull width, orbit length, ulna length, femur length and tibia length) shows a distinct pattern, with the three genera separated in morphospace, especially along PC-1 (Fig. 9D). Furthermore, neither the French or German specimens show equivalent distributions to each other, which might be expected if each basin records the same taxon along one growth series. The two *Atoposaurus* species are distinguished by the PC-1 (94.1% variance). *Alligatorellus* species are weakly distinguished from each other by PC-2 (2.8% variance), but show almost no variation in PC-1. Whereas the two *Alligatorium* species are strongly distinguished from each other on PC-2, they are closely united by PC-1 (Fig. 9D). As such, we find no convincing uni-directional evidence that *Atoposaurus*, *Alligatorellus*, and *Alligatorium* form a single growth series of one species. Although we cannot fully preclude the possibility

that *Atoposaurus* represents an immature specimen of either *Alligatorellus* or *Alligatorium*, below we discuss other anatomical features that support its taxonomic validity.

*Atoposaurus* is unique among all unequivocal atoposaurids in lacking osteodermal armour. *Schwarz-Wings et al. (2011)* suggested that this might merely be a taphonomic artefact; however, preservational selectivity of this nature seems unlikely given that there is no clear reason why similarly ossified parts of the skeleton would undergo differential preservation. Combined with its diminutive size, the absence of any cranial sculpting, and lack of calcified palpebrals, the absence of osteoderms might suggest a juvenile status. Furthermore, *Atoposaurus* looks superficially similar to a juvenile specimen of the alligatoroid *Diplocynodon* from Messel (Eocene; *Delfino & Sánchez-Villagra, 2010*, Fig. 3A), in terms of the relatively long caudal vertebral series, large orbits, lack of ossification of the dermal armour, and proportionally short skull. As such, *Atoposaurus* superficially takes on the appearance of more advanced eusuchians, while retaining paedomorphic characteristics (e.g., the proportionally large orbits). In crocodylians, the initiation of osteoderm ossification is usually substantially delayed relative to skeletal ossification (*Vickaryous & Hall, 2008*), so it is difficult to infer a more accurate ontogenetic age for *Atoposaurus* specimens based solely on a lack of osteoderms. However, there are additional morphological aspects that demonstrate that *Atoposaurus* might not be a juvenile.

Similar to most other atoposaurids, *Atoposaurus* has a relatively short, low, acute, triangular skull. However, as with some theropod dinosaurs, the extant crocodylian *Osteolaemus*, and perhaps even the alligatoroid *Alligator*, shortening of the rostral region may be a paedomorphic feature, with the morphology similar to juveniles and sub-adult specimens of *Melanosuchus* (the black caiman) (*Foth, 2013*). A platyrostral skull is also known in basal eusuchians such as *Iharkatosuchus maxakii* (*Osi, Clark & Weishampel, 2007*), and is distinct from the majority of contemporaneous crocodylomorphs, including goniopholidids and thalattosuchians. Heterochrony in crocodylomorphs may be directly related to body size or diet, as atoposaurid species and *Osteolaemus* are relatively small forms that occupy distinctive ecologies. However, patterns of heterochrony, particularly relating to paedomorphosis, in 'dwarfed' species are currently poorly understood in crocodylomorphs, but could be responsible, at least in part, for the lack of osteoderm ossification in *Atoposaurus*.

The degree of suturing between the vertebral centrum and neural arch provides ontogenetic information (*Mook, 1933*; *Brochu, 1996*). Closure of cervical sutures is a consistent indicator of morphological maturity, and is known in basal crocodylomorphs (e.g., thalattosuchians; *Delfino & Dal Sasso, 2006*) and advanced eusuchians (*Brochu, 1996*). In *Atoposaurus jourdani*, the neural arches are fused to the centra (MNHN 15680; J Tennant, pers. obs., 2013), which implies that this specimen represents a more mature growth stage despite the size of the individual. Furthermore, it is interesting to note that other putative atoposaurids of diminutive size, such as the 160 mm long *Karatausuchus* (*Efimov, 1976*), also lack osteoderms, suggesting that osteoderm development might be positively correlated with body size in atoposaurids. We therefore suggest that *Atoposaurus* represents an extreme case of dwarfism.

*Alligatorium* is currently composed of three species: *A. meyeri* from Cerin, France (*Vidal, 1915*), and *A. franconicum* (*Ammon, 1906*) and *A. paintenense* (*Kuhn, 1961*) from Painten, central Bavaria, Germany. However, based on the figures and descriptions provided by *Wellnhofer (1971)*, *A. franconicum* (an articulated hindlimb and pelvic girdle) cannot be distinguished from *A. paintenense* (a near-complete, articulated skeleton), aside from slight differences in femur-to-tibia length proportions. Given that both specimens are from the same locality, we tentatively conclude that they do not represent distinct species, and regard *A. paintenense* (*Kuhn, 1961*) as synonymous with *A. franconicum* (*Ammon, 1906*), pending the relocation of the type material and/or discovery of new material. The type specimen of *A. paintenense* is clearly distinct from *A. meyeri* and both species of *Alligatorellus*, based on its more longirostrine snout, and dorsal osteoderms which each possess a longitudinal keel and an anterolateral hook (*Wellnhofer, 1971*). *Alligatorium meyeri* can be distinguished from *Alligatorellus* based on the absence of a longitudinal keel on all osteoderms in the latter taxon, as well as disparity in the cranial sculpting between the two taxa. As well as differing from *Alligatorellus* and *Alligatorium* in terms of the absence of osteoderms, *Atoposaurus* can also be distinguished via caudal osteoderm counts, with ten less caudal vertebrae in the latter taxon (the number is unknown for *Alligatorium*). A higher number of caudal vertebrae in *Atoposaurus* is additional evidence that this taxon is not an immature individual of at least *Alligatorellus*, given that we would not expect an individual to lose vertebrae with increasing maturity. In summary, we retain *Alligatorellus*, *Alligatorium* and *Atoposaurus* as distinct atoposaurid genera, with each genus comprising a valid French and German species.

## European atoposaurid diversity

The Late Jurassic–Early Cretaceous of Europe records high atoposaurid diversity, comprising the multispecific genera *Alligatorellus*, *Alligatorium*, *Atoposaurus* and *Theriosuchus*, as well as *Montsecosuchus depereti* (*Gervais, 1871*; *Owen, 1879*; *Wellnhofer, 1971*; *Buscalioni & Sanz, 1990a*; *Brinkmann, 1992*; *Schwarz & Salisbury, 2005*). Currently valid European species of *Theriosuchus* include: *T. guimarotae* from the Kimmeridgian of Portugal (*Schwarz & Salisbury, 2005*); *T. pusillus* from the Berriasian of England (*Owen, 1879*; *Salisbury, 2002*); *T. ibericus* from the Barremian of Spain (*Brinkmann, 1989*; *Brinkmann, 1992*); and *T. sympiestodon* from the Maastrichtian of Romania (*Martin, Rabi & Csiki, 2010*; *Martin et al., 2014*). However, support for the monophyly of these species of *Theriosuchus* has yet to be adequately evaluated. Such evaluation is particularly required in view of the spatiotemporal distribution of the genus as currently understood, which spans some 90 million years and includes a putative Asian occurrence (*T. grandinaris*; *Lauprasert et al., 2011*).

Along with these relatively well-known species, there is a host of European material ascribed to *Theriosuchus* sp. from: the Kimmeridgian of northwest Germany (*Thies, Windolf & Mudroch, 1997*; *Karl et al., 2006*); the Berriasian of Scandinavia (*Schwarz-Wings, Rees & Lindgren, 2009*); the Berriasian of Charente, France (*Poueche, Mazin & Billon-Bruyat, 2006*); the Berriasian–Valanginian of northern Germany (*Hornung, 2013*); the Valanginian–Barremian of England (*Buffetaut, 1983*); and the Hauterivian–Barremian

of Iberia (*Buscalioni & Sanz, 1984*; *Buscalioni & Sanz, 1987b*; *Ruiz-Omenaca et al., 2004*; *Buscalioni et al., 2008*; *Canudo et al., 2010*). Indeterminate atoposaurid remains from the Cenomanian of France (*Vullo & Néraudeau, 2008*), and mid-Coniacian Kaiparowits Formation of Utah, US (*Eaton et al., 1999*), as well as *Theriosuchus*-like teeth from the Santonian of Hungary (*Ösi et al., 2012*) and the Upper Campanian-Maastrichtian of Portugal (*Galton, 1996*), bridge the temporal gap between these Late Jurassic–Early Cretaceous atoposaurid remains and *Theriosuchus sympiestodon* from the latest Cretaceous of Romania (*Martin, Rabi & Csiki, 2010*; *Martin et al., 2014*). They also hint at a cryptic diversity of mid-Late Cretaceous atoposaurids, as well as their presence in North America. Additionally, tracks from the Kimmeridgian of Asturias, Spain (*Avanzini, Piñuela & Garcia-Ramos, 2010*), might be attributable to atoposaurids, extending their Late Jurassic geographic range. The taxonomic utility of crocodyliform teeth clearly requires further investigation, and may help to resolve scenarios where multiple, but clearly distinct, tooth morphotypes are present in the same locality (e.g., the presence of atoposaurid-like teeth alongside *Theriosuchus* throughout the late Berriasian-early Aptian Wealden Group, UK; *Sweetman, 2011*).

It is possible that the high diversity of Late Jurassic to Early Cretaceous European atoposaurids is related to the island archipelago system that existed during this time (Fig. 1), with epicontinental seas driven by fluctuating highstand sea levels (*Ziegler, 1988*; *Schwarz & Salisbury, 2005*; *Miller et al., 2005*). The separation of areas (e.g., basins in present day Cerin and Bavaria) might have led to allopatric speciation, evidenced by closely related species found in each region (i.e., *Alligatorellus beaumonti*, *Alligatorium meyeri* and *Atoposaurus jourdani* in Cerin, and *Alligatorellus bavaricus*, *Alligatorium franconicum* and *Atoposaurus oberndorferi* in Bavaria). The small body size of atoposaurids in general might also be explained by these environmental conditions, via ecological partitioning with other contemporary crocodyliforms, including thalattosuchians and goniopholidids. There is potentially evidence for niche partitioning in the Early Cretaceous of western Europe, when comparably small-bodied bernissartiid crocodylomorphs lived alongside *Theriosuchus*. Whereas both groups had a heterodont dentition, bernissartiids also possessed tribodont teeth, suited to a durophagous or conchifragous diet (*Buffetaut & Ford, 1979*; *Sweetman, Pedreira-Segade & Vidovic, in press*). This dietary partitioning might have been key to two otherwise similar groups living side-by-side. It is also possible that the small body size of atoposaurids (and potentially bernissartiids) reflects insular dwarfism driven by a sea level-driven reduction in range size, as also proposed for the contemporaneous Late Jurassic German sauropod dinosaur *Europasaurus* (*Sander et al., 2006*; *Marpmann et al., in press*). This reasoning is also supported by the persistence of atoposaurids into the Maastrichtian as part of an assemblage of insular island dwarfs in a range of environments and localities, including the Haţeg Basin of Romania (*Benton et al., 2010*; *Csiki & Benton, 2010*; *Martin, Rabi & Csiki, 2010*; *Martin et al., 2014*). Dwarf crocodiles are also known from the Quaternary of the Aldabara Atoll (western Indian Ocean), with *Aldabrachampsus dilophus* (*Brochu, 2006*) indicating that island dwarfism in

crocodylomorphs might not be an uncommon feature. The existence of three sympatric lineages of the dwarf crocodile *Osteolaemus* in present day western Africa (*Eaton et al., 2009*; *Shirley et al., 2013*) also supports the idea that atoposaurids could similarly have had multiple co-existing lineages, such as that seen in the French and German basins.

Currently, testing of these hypotheses is limited as a result of the small number of localities preserving atoposaurids. To support the hypothesis of insular dwarfism, basal members of Atoposauridae should be expected to be much larger than these Late Jurassic European forms; however, we will only be able to demonstrate this with the discovery of well preserved, stratigraphically older forms, from non-island archipelago settings.

## CONCLUSIONS

We have presented a new description of a Late Jurassic German atoposaurid specimen previously referred to a subspecies of *Alligatorellus beaumonti*, otherwise known only from coeval deposits in France. We recognise it as a distinct species of *Alligatorellus*, based on numerous features across the skeleton, and re-rank it as *Alligatorellus bavaricus*. Emended diagnoses are provided for the genus, as well as the French and German species. Comparisons with contemporaneous atoposaurids support the validity of *Atoposaurus* and *Alligatorium*, alongside *Alligatorellus*, with a species of each genus present in Late Jurassic basins in both France and Germany, providing evidence for sea level-driven allopatric speciation.

**Institutional Abbreviations**

| | |
|---|---|
| **LMU** | Ludwig-Maximilians Universitat, Bayerische Staatssammlung für Paläontologie und Geologie, München, Germany |
| **MfN** | Museum für Naturkunde, Berlin, Germany |
| **MNHN** | Muséum National d'Histoire Naturelle, Centre de Conservacion, Lyon, France |
| **NHMUK** | Natural History Museum, London, UK |
| **TMH** | Teylers Museum, Haarlem, The Netherlands. |

**Anatomical Abbreviations**

| | |
|---|---|
| **Af** | antorbital fenestra |
| **Cav** | caudal vertebra |
| **Cev** | cervical vertebra |
| **Ch** | chevron |
| **C** | coracoid |
| **Co** | caudal osteoderm |
| **Do** | dorsal osteoderm |
| **Dov** | dorsal vertebra |
| **Dr** | dorsal ridge |
| **Ds** | dermal sculpting |

| | |
|---|---|
| **En** | external nares |
| **Fi** | fibula |
| **Fr** | frontal |
| **Hu** | humerus |
| **Il** | ilium |
| **Is** | ischium |
| **Ju** | jugal |
| **La** | lacrimal |
| **Ltf** | lateral temporal fenestra |
| **Ma** | manus |
| **Man** | mandible |
| **Max** | maxilla |
| **Mp** | manual phalanx |
| **MT** | metatarsal |
| **Na** | nasal |
| **No** | nuchal osteoderm |
| **Or** | orbit |
| **Pa** | parietal |
| **Pal** | palatine |
| **Pe** | pes |
| **Pmn** | premaxilla-maxilla notch |
| **Pmx** | premaxilla |
| **Po** | postorbital |
| **Pp** | pedal phalanx |
| **Pu** | pubis |
| **Ra** | radius |
| **Qj** | quadratojugal |
| **Qu** | quadrate |
| **Rad** | radiale |
| **Ri** | rib |
| **Sc** | scapula |
| **So** | sacral osteoderm |
| **Sof** | suborbital fenestra |
| **Sq** | squamosal |
| **Stf** | supratemporal fenestra |
| **Sym** | symphysis |
| **Ti** | tibia |
| **Ul** | ulna |
| **Uln** | ulnare |
| **Up** | ungual phalanx |

## ACKNOWLEDGEMENTS

We are grateful to Oliver Rauhut (LMU), Didier Berthet (MNHN), Lorna Steel (NHMUK), Herman Voogd (TMH) and Daniela Schwarz-Wings (MfN) for providing access to specimens in their care. Comments from Trevor Valle and Joseph Hancock greatly improved an earlier draft of this manuscript, and we are especially grateful to Mark Young and Steve Sweetman for their reviews that substantially enhanced this study. Additionally, we would like to thank Matt Wedel for his input as Editor.

### Funding

JPT is funded by a NERC PhD studentship (EATAS G013 13). PDM is funded by an Imperial College Junior Research Fellowship. The funders had no role in study design, data collection and analysis, decision to publish, or preparation of the manuscript.

### Grant Disclosures

The following grant information was disclosed by the authors:
NERC PhD studentship: EATAS G013 13.
Imperial College Junior Research Fellowship.

### Competing Interests

The authors declare there are no competing interests.

### Author Contributions

- Jonathan P. Tennant and Philip D. Mannion conceived and designed the experiments, performed the experiments, analyzed the data, contributed reagents/materials/analysis tools, wrote the paper, prepared figures and/or tables, reviewed drafts of the paper.

### New Species Registration

The following information was supplied regarding the registration of a newly described species:

zoobank.org:pub:B7CC4367-4203-4AED-8C30-2D7E4E71665D.

### Supplemental Information

Supplemental information for this article can be found online at http://dx.doi.org/10.7717/peerj.599#supplemental-information.

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
