# Peer review of "Revision of the Late Jurassic crocodyliform Alligatorellus, and evidence for allopatric speciation driving high diversity in western European atoposaurids"

_PeerJ, doi:10.7717/peerj.599_

## Round 0.1 · original submission · Major Revisions

· Academic Editor

Major Revisions

After looking over the manuscript and both sets of reviews, I find the comments by the reviewers to be reasonable, and I ask that you address their concerns wherever possible in revising the manuscript. A few points demand particular attention:

1. Both reviewers requested additional illustrations to clarify the phylogenetically informative characters. I know that re-photographing material and preparing new illustrations can be fairly labor-intensive, but I see this as an opportunity to not only address the reviewers' concerns but also to add lasting value to the work.

2. Both reviewers found parts of the discussion problematic, in particular those sections dealing with insular dwarfism and cryptic diversity. Although I find your arguments plausible, both reviewers raise important questions that I encourage you to address in your revision.

3. Please pay special attention to the comments of Reviewer 2 regarding taxonomy, taxonomic authorities, and museum specimen numbers. In cases where specimens have been referred to in different ways in previous publications, either because of renumbering or changes in the preferred format of specimen numbers, it may be useful to provide a table listing earlier and later specimen numbers.

·

Basic reporting

In general, this is a carefully considered and well-argued contribution. However, the MS could be shortened somewhat by removal of some unnecessary repetition. Also, the wording can be tightened, and altered in certain places to avoid language that does not comply with generally accepted scientific standards. I also found some of the language very awkward and in places confusing and have have made numerous suggestions for changes on the annotated MS.

Anatomical descriptions are refreshingly clear and concise. But there are one or two examples of ambiguous wording. This should be altered where indicated and figures provided (see below).

No description of characters defining Atoposauridae is provided and differential diagnoses are lacking. I do not feel the MS is complete without them.

References are in good order except where indicated but the authors need to check for compliance with journal style requirements, mostly in the case of references to articles and chapters appearing in books (which I have not done in the case of any aspect of the MS). However, Fiorillo 1999 is cited in the text but is not included in the list of references and Buscalioni et al. 2011 appears in the list but is not cited in the text.

So far as figures are concerned, Fig. 1 shows dotted lines in black for Theriosuchus sp. and Atoposauridae indet. No explanation of the geographical occurrence(s) is provided. Furthermore, I suspect that indeterminate isolated teeth representing atoposaurids are known from deposits throughout their spatiotemporal range. E.g. indeterminate (or possibly undetermined) teeth representing a small atoposaurid are abundant among microvertebrate remains recovered from the Barremian Wessex Formation (Sweetman. 2011. Text-fig. 16.4E,F p. 201, in Batten (ed.) Palaeontological Association Field Guide to Fossils 14. English Wealden Fossils. Pal. Ass. London).

As mentioned in an annotation, I think it would be better to place the anatomical annotations shown in Fig. 2 below the institutional abbreviations. With regard to the former, the authors should give thought to their format and the use of capital letters, e.g. AF is used for Antorbital fenestra whereas Cav is used for caudal vertebra. This looks untidy to my eye. Perhaps use juxtaposed capitals only for osteoderms?

Figures of described specimens, so far as they go, are satisfactory but the authors need to provide detailed drawings of many of the elements they describe (unless these were provided by Wellnhofer (1971), in which case reference to his figures should be provided in the text where appropriate). Additional figures must include skull diagrams to aid comparisons between Alligatorellus beaumonti and A. bavaricus.

There was no reference to Fig. 6 in the text. I have made one suggestion but, if it is retained, reference to it should be made elsewhere. I find comparison between A and B confusing. Localities 5 and 6 are shown in the same place on each map but B shows the Iberian Peninsula as it was before the current configuration was achieved. Also, what does the diagonal line towards the bottom in B represent? Paleaogeography did not change much between the Late Jurassic and Early Cretaceous and the map provided is simplistic and schematic so I think it would be useful to include early Cretaceous occurrences (indicated by a different coloured dot) as these are discussed in the text.

In the case of all Figs. capital letters are used do designate elements in each whereas lowercase letters are used in the captions. Also, in Fig. 6 bold text is not required. The Captions should be changed.

Experimental design

NA

Validity of the findings

I agree with the authors’ re-evaluation of the taxa discussed. I also agree with their comments concerning allopatric speciation, not least because a recent study has confirmed that very short geographical separations can result in speciation (Sweetman et al. In press. A new bernissartiid crocodyliform from the Lower Cretaceous Wessex Formation (Wealden Group, Barremian) of the Isle of Wight, southern England. Acta Pal. Pol.). However, I strongly disagree with those concerning insular dwarfism. There is no evidence for this so far as crocodyliforms in this part of Laurasia are concerned.

Looking at the Early Cretaceous (Berriasian – Early Aptian, a time span of at least 20 million years during which the area under consideration still comprised an island archipelago) diminutive crocodyliforms such as atoposaurids coexisted with large forms, e.g. goniopholids. Early Cretaceous atoposaurids appear to have been about and remain the same size as they were during the Late Jurassic and there is no sign of a reduction in size of large taxa. It seems much more likely to me that size difference relates to adaptations between taxa to exploit different trophic resources in the environments in which they coexisted. Furthermore, the authors fail to but should make mention of bernissartiid crocodyliforms. These are of a similar size to coexisting atoposaurids but with a unique dentition indicating further environmental partitioning among small crocodyliforms. A couple of words concerning the small size of atoposaurids is appropriate but the long discussion about dwarfism should be removed.

Additional comments

In addition to the above please refer to the annotated MS and advise the editor if you would like this as a Word document. Please also note that the title and abstract page was tacked onto the top of the main part of the MS as it was not part of the source file. Two deleted words near the bottom of the abstract (three and the) are not shown as such in the markup.

Please note that I am somewhat dyslexic. I may have made typographical errors in some of my comments and may not have seen others in the original MS.

·

Basic reporting

All relevant comments are in my general view.

Experimental design

No Comments

Validity of the findings

I cannot determine the validity of the sub-specific elevation based on the current figures. (Comment 1 of my review.)

Additional comments

1. Based on the photographs provided I cannot determine whether the two sub-species of Allgatorellus beaumonti should be split into two separate taxa. Can the authors provide more photographs? Especially of the taxonomically important characters that vary between the two sub-species?
2. Depending on the journal-style, the 19th Century author Meyer can be written as von Meyer.
3. It is nice to have all the taxonomic history of Atoposaurus in the introduction, but as this manuscripts focus is on Alligatorellus is it really necessary? Would it not be better to hold this back to another manuscript? It does make the introduction taxonomically 'dense'. Perhaps it would be better to make the introduction more succinct, and simply cover: 1) general atoposaurid background, 2) history of Alligatoreluus and 3) the why this matters/what you intend to do. At the moment it reads like the authors are trying to 'shoe-horn' in everything they know about atoposaurids into the introduction.
4. I would recommend the authors double check their Muséum national d’Histoire naturelle (note both national and naturelle are lower case) specimen numbers are correct. The paléontologie collections in the MNHN have been in the process of changing their specimen numbering system.
5. The Natural History Museum London specimen numbers are not correctly written. Theriosuchus pusillus is written as: NHMUK 48330. The current citation system used by the NHMUK would give that as: NHMUK PV OR48330. If the authors have any doubt about specimen numbers, they should double check them with the NHMUK.
6. Under the ICZN Code Alligatorellus bavaricus is not a new species. The nominal author is still Wellnhofer (1971). All the authors have done is elevate the sub-specific epithet up one taxonomic rank to a specific epithet. Many sub-generic names proposed in the 19th Century (such as those proposed by Cuvier and Eudes-Deslonchamps) for fossil material are now used as generic names. The authorship of those names is still the nominal author of the sub-generic name, not the one who raised it a taxonomic level. I would recommend the authors read Article 50.3 of the ICZN Code. Moreover, it is useful for all taxonomists to be familiar with the contents of the Code to prevent taxonomic confusion.
7. The nominal author of Crocodylomorpha is Hay (1930). Walker (1970) is the currently used definitional authority.
8. The umlaut over the 'a' in Eichstätt is missing throughout the manuscript.
9. The "heterogeneous degree of cranial sculpting" mentioned in the description is likely to be ontogenetically controlled (as well as possibly differing phylogenetically).
10. In the discussion there are statements made that lack references to support them.
11. In the discussion it can be somewhat hard to track which genus is being discussed as all three Franco-German genera begin with 'A'. Rather than stating A. meyeri and A. beaumonti, it might be best to use 'At.', 'Am.' and 'As.' for disambiguation.
12. I have to say that the ratios presented make a good case for Atoposaurus being a juvenile of either Alligatorium or Alligatorellus. Atoposaurus has: 1) a short snout to skull length ratio (a juvenile characteristic throughout long-snouted vertebrates), 2) narrow skull width to skull length ratio (which in extant crocodylians varies ontogenetically and through sexually dimorphism), and 3) variation in long-bone length ratios (which again varies ontogenetically in extant crocodylians). This section of the discussion will require a major reappraisal, and would benefit enormously with comparisons to extant crocodylian ontogenetic variation.
13. The discussion section: "European Atoposaurid Diversity" feels like a non-sequitur, and again seems like someone adding in what they know about atoposaurids rather than maintaining a cohesive structure (especially the first two paragraphs). Moreover, as I think that Alligatorium, Alligatorellus and Atoposaurus are unlikely to all be valid genera, discussion on 'high cryptic diversity' seems premature. And are modern dwarf crocodylians really that conservative, or is that more of a straw-man argument? I suspect the latter, as morphological differences typically are found.
14. There are references in the reference list that are not cited in the paper.
15. In Figure 6, localities 5 and 6 are labelled on what would be Algeria, not Portugal as they should be. Locality 8 is too far north.

---

## Round 0.2 · accepted · Accept

· Academic Editor

Accept

Thank you for your attention to detail in revising the manuscript and writing your response letter. After considering the reviews, your responses, and the revised manuscript, I am happy to accept your manuscript for publication in PeerJ.

Although it is of course up to you whether to publish the review history alongside your manuscript, I think there would be much to gain by doing so. Both the reviews and your responses to them were concise and constructive, and I think it's a great example of the review process working as it should.